# RETHINKING REASONING WITH MASKED DIFFUSION MODELS

## ABSTRACT

Masked diffusion language models (MDLMs) are trained to in-fill positions in randomly masked sequences, in contrast to traditional next-token prediction (NTP) models. Discussions around MDLMs focus on two benefits: (1) any-order decoding and 2) multi-token decoding. However, we observe that for math and coding tasks, any-order algorithms often underperform or behave similarly to *left-to-right sampling*, and standard multi-token decoding significantly degrades performance. At inference time, MDLMs compute the conditional distribution of all masked positions. A natural question is: *How can we justify this additional compute when left-to-right one-token-at-a-time decoding is on par with any-order decoding algorithms?* These findings warrant rethinking how MDLMs are utilized. First, we propose *reasoning-as-infilling*. By using MDLMs to infill a reasoning template, we can structure outputs and distinguish between reasoning and answer tokens. In turn, this enables measuring answer uncertainty *during* reasoning, and early exits when the model converges on an answer. Next, given an answer, *reasoning-as-infilling* enables sampling from the MDLM posterior over reasoning traces *conditioned on the answer*, providing a new source of high-quality data for post-training. On GSM8k, we observe that fine-tuning LLaDA-8B Base on its posterior reasoning traces provides a performance boost on par with fine-tuning on human-written reasoning traces. Additionally, given an answer, reasoning-as-infilling provides a method for scoring the correctness of the reasoning process at intermediate steps, without requiring expensive rollouts or an external model. Second, we propose multi-token entropy decoding (MED), a simple adaptive sampler that minimizes the error incurred by decoding positions in parallel based on the conditional entropies of those positions. MED preserves performance across benchmarks and leads to $2.7\times$ fewer steps. Combined with early exits, MED leads to a $3.3\times$ speed-up on GSM8k with a minimal ($0.1\%$) effect on accuracy. Our work demonstrates that the training objective and compute used by MDLMs unlock many new possibilities for inference and post-training methods.

## 1 INTRODUCTION

The current dominant approach for language modeling is based on next-token prediction (NTP) training. NTP language models learn the conditional distribution of the *next token* given the previous tokens in a sequence (Shannon, 1951; Radford et al., 2019). The resulting language model is sampled auto-regressively left-to-right, one token at a time. Recent work proposes MDLMs (Austin et al., 2021; Sahoo et al., 2024; Shi et al., 2024) as an alternative to NTP models. MDLMs are trained to in-fill sequences with randomly masked positions. The resulting model learns the distribution $p_\theta(x^i | \mathbf{x}_{\text{UN-MASKED}})$ at every masked position $i$.

While modeling all masked positions requires additional effort, MDLMs have several potential benefits, such as parallel token decoding (Sahoo et al., 2024; 2025), and flexible decoding orders (Kim et al., 2025) that lead to significant improvements on logic puzzles, such as Sudoku. Additionally, Bachmann & Nagarajan (2024); Prabhudesai et al. (2025) show that multi-token prediction objectives can achieve better likelihoods and accuracy on tasks, and access to the distribution and samples from masked positions supports controllable generation (Schiff et al., 2024; Singhal et al., 2025).

In our work, we first examine two benefits of MDLMs: any-order and multi-token decoding, on mathematical reasoning and coding benchmarks. Despite the flexibility enabled by MDLMs, we

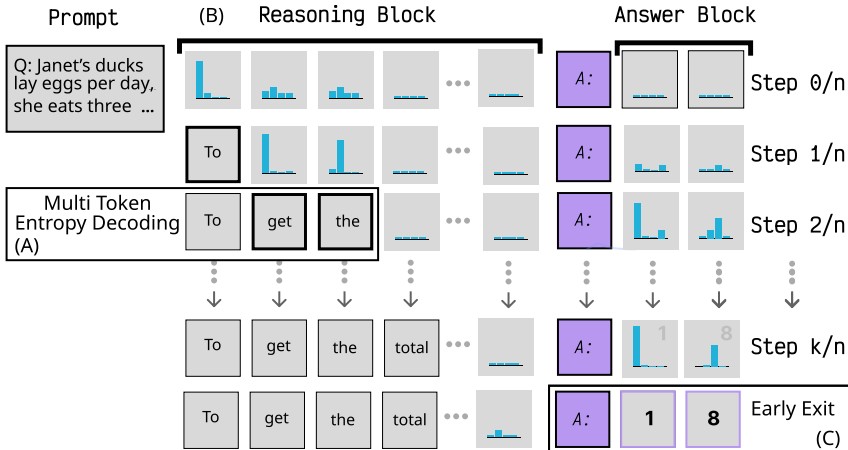

**Figure 1:** MDLMs learn the conditional distributions at each masked token position. A) We reframe reasoning as infilling a prompted reasoning template, which enables directly modeling answer token probabilities *during* reasoning. This provides several benefits, like B) enabling early exits or *post-hoc* reasoning given a pre-filled answer. C) We also utilize the entropy of these distributions to adaptively set the number of tokens decoded at each step.

observe that decoding one token in a left-to-right order, identically to an NTP model, is a strong decoding choice for MDLM models. Even decoding just two tokens in parallel substantially reduces performance on popular benchmarks. These findings raise questions about the substantial extra compute MDLMs spend to model the distribution of all masked positions. In this work, we show how this compute can be made *useful*. We demonstrate that the access that MDLMs provide to the conditional distributions of all masked positions, and their ability to in-fill, unlocks new sampling and post-training capabilities that are not readily available for NTP models.

First, we demonstrate that the ability of MDLM to in-fill opens up new model prompting paradigms. In this work, we propose prompting-as-infilling, where we add user-specified contexts in multiple positions, not just the beginning of the sequence, unlike NTP models. Specifically, we consider **reasoning-as-infilling**. Here we pre-fill an explicit reasoning template, with specific reasoning and answer positions (see fig. 1). This enables sampling reasoning traces conditioned on a reasoning budget and format. We demonstrate that the in-filled template provides many advantages. By explicitly distinguishing token answer positions, we can make use of the conditional distributions of the masked positions provided by MDLMs to measure the uncertainty of the answer *while reasoning*. In turn, this enables **early exits** once the model converges on an answer, reducing inference costs. For instance, on GSM8k this leads to $24\%$ fewer function calls with no degradation in accuracy.

Reasoning-as-infilling has consequences for analyzing model behavior and improving performance. Given access to an answer, we can sample from the MDLM's posterior distribution of reasoning traces conditioned on the answer, $p_\theta(\mathbf{r} \mid \mathbf{c}, \mathbf{a})$. This easy sampling from the posterior in MDLMs enables generating high-quality *post-hoc* reasoning traces for use in model fine-tuning.

Next, we revisit multi-token decoding. Decoding multiple positions in a single step results in samples that are not from the MDLM's learned distribution, as typically the joint distribution and factorized distributions do not align, $p_\theta(x^i, x^j \mid \mathbf{x}_{\text{UN-MASKED}}) \neq p_\theta(x^i \mid \mathbf{x}_{\text{UN-MASKED}})p_\theta(x^j \mid \mathbf{x}_{\text{UN-MASKED}})$. However, by making use of the entropy of the masked positions to inform decoding, we can control how much multi-token decoding deviates from single token sampling. We propose Multi-token Entropy Decoding (MED), an **adaptive multi-token decoder** that decodes multiple positions only if the conditional entropy of the additional positions falls below a specified threshold. We find that MED leads to $2\text{-}3\times$ fewer function calls, with a minor or no drop in performance.

**Contributions.** In this paper, we:

- Evaluate MDLM models, such as Dream (Wu et al., 2025b) and LLaDA (Nie et al., 2025), on several tasks and find that the any-order sampling capability of MDLM provides limited benefits on

coding and mathematical reasoning benchmarks, and that standard multi-token decoding degrades performance.

- Introduce reasoning-as-infilling for MDLMs, which leverages their infilling capabilities. We then show that distinguishing reasoning and answer tokens can provide several benefits, such as:

  - *Early exits*, where if the model is certain about the answer, we then skip the remaining reasoning steps. This leads to a $3.3\times$ speed when combined with multi-token entropy decoding (MED).

  - *Post-hoc reasoning*, where given question-answer pairs, we generate reasoning traces conditioned on the answer. On the GSM8k dataset, we find that supervised fine-tuning on these reasoning traces can improve the model more than supervised fine-tuning on the *human-annotated* GSM8k reasoning traces.

  - *Scoring reasoning traces*, where given an answer, we can score the reasoning process for correctness at intermediate steps using the distributions of the answer block, without an external verifier or roll-outs. These scores correlate with whether the reasoning steps lead to a correct answer.

- Propose MED, an adaptive sampler that provides a 2-3$\times$ speed-up, without any loss in performance on math and coding benchmarks.

## 2 RELATED WORK

**Multi-Token Prediction and Speculative Decoding.** Gloeckle et al. (2024) show that models trained with the multi-token objective can enable parallel multi-token decoding, or *speculative decoding*, without making use of another model. However, unlike MDLMs, Gloeckle et al. (2024) limit to predicting the next $2, 4$ tokens. Several other works (Leviathan et al., 2023; Chen et al., 2023) show that using smaller draft models for generation and then rejection sampling can also enable parallel decoding with NTP models. However, MDLMs offer many possibilities beyond left-to-right parallel decoding, such as in-filling, and error correction through re-masking of unmasked tokens (Wang et al., 2025). Israel et al. (2025) propose an adaptive multi-token decoder which samples from the product of an NTP and MDLM model. Unlike MED, their approach relies on rejection sampling based on an external NTP model.

Ben-Hamu et al. (2025) propose entropy-bound (EB) sampler, an adaptive multi-token decoder for MDLMs, which similar to MED controls the error incurred by multi-token decoding. EB sampler adds multiple positions to unmask based on the difference between the sum of the positions added and the maximum entropy until the difference exceeds a specified threshold $\gamma$. MED unmasks positions based on the individual entropies rather than thresholding based on the sum. Wu et al. (2025a) propose accelerating inference with MDLMs using KV-caching and parallel decoding. Similar to Ben-Hamu et al. (2025), they propose an adaptive greedy strategy, instead decoding tokens with confidences above a fixed probability threshold, unlike MED, which thresholds based on entropy. Additionally, the method proposed in Wu et al. (2025a) is designed to accelerate $\arg\max$ sampling from MDLMs, whereas MED is compatible with inference-time steering methods and post-training methods that require multiple samples per prompt, such as RLOO (Ahmadian et al., 2024) and GRPO (Shao et al., 2024; Zhao et al., 2025). We include adaptive sampler comparisons in section A.

**Post-hoc Reasoning.** Zelikman et al. (2022) proposes fine-tuning language models on reasoning traces that are generated conditioned on a correct answer. Phan et al. (2023); Ruan et al. (2025) propose fine-tuning a model on samples from approximations of the posterior $p_\theta(\mathbf{r} \mid \mathbf{c}, \mathbf{a})$. In contrast, MDLMs enable exact sampling from the posterior of the reasoning traces given the answer by simply in-filling the answer in the answer block provided in the reasoning-as-infilling framework.

## 3 MASKED DIFFUSION LANGUAGE MODELS

MDLMs (Sohl-Dickstein et al., 2015; Devlin et al., 2019; Austin et al., 2021; Sahoo et al., 2024; Shi et al., 2024) are a class of generative models for modeling discrete data $\mathbf{x} \sim q_{\text{data}}$, where $\mathbf{x} = (x^1, x^2, \ldots, x^L)$ and each position $x^i$ takes values in a finite vocabulary $\mathcal{V}$. For training the model $p_\theta(\mathbf{x} \mid \mathbf{c})$, the sequence $\mathbf{x} \sim q_{\text{data}}$ is masked randomly and the model learns to predict the

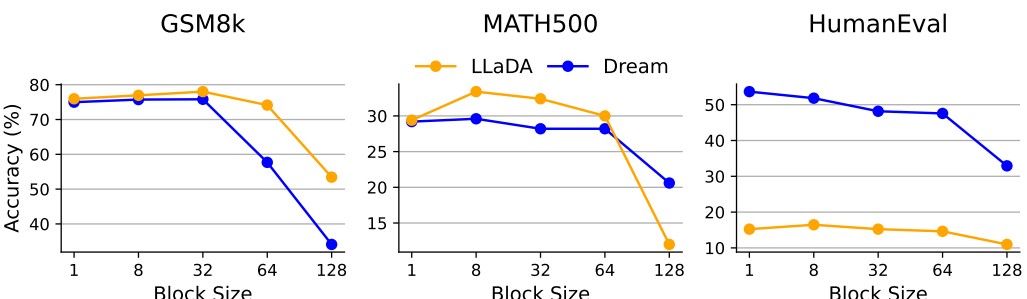

**Figure 2: Left-to-right sampling with MDLMs is a competitive sampling algorithm for reasoning and coding.** When performing entropy decoding (Ye et al., 2025b), we observe that full any-order sampling results in poor performance on all tasks but Sudoku. Left-to-right block decoding is required to make any-order sampling performant, and left-to-right sampling (block size = 1) is always within a few percent of the best configuration. We also observe in table 7 in section B that performant block any-order configurations sample a large portion of tokens left-to-right. We consider sequences of length 128.

distributions of the masked positions for a fixed length sequence by maximizing:

$$\mathcal{L}(\mathbf{x}, \mathbf{c}, \theta) := \mathbf{E}_{\text{MASKED-SET} \sim U} \sum_{j \in \text{MASKED-SET}} \log p_\theta(x^j \mid \mathbf{x}_{\text{UN-MASKED}}, \mathbf{c}) \qquad (1)$$

where MASKED-SET is a set of randomly masked positions in $\{1, 2, \ldots, L\}$. Sampling from an MDLM is performed by iteratively un-masking positions. Notably, an MDLM can also be viewed as any-order auto-regressive model (Uria et al., 2014; Ou et al., 2024), where given a decoding order $\mathbf{o} = [o_1, \ldots, o_L]$ with $o_j \in \{1, 2, \ldots, L\}$, the sampling model can be defined as:

$$p_\theta(\mathbf{x} \mid \mathbf{c}, \mathbf{o}) = \prod_{j=0}^{L-1} p_\theta(x^{o(j)} \mid \mathbf{x}_{o(<j)}, \mathbf{c}) \qquad (2)$$

where $o_j$ refers to the position decoded at step $j$ and $\mathbf{x}_{o(<j)}$ refers to all positions decoded prior to step $j$. Additionally, block-sampling (Sahoo et al., 2024; Nie et al., 2025; Arriola et al., 2025) approaches define a left-to-right sequence of fixed length blocks and decode within each block in an arbitrary order.

### 3.1 PRELIMINARY OBSERVATIONS

In our work, we first examine two purported benefits of MDLMs: *any-order* and *multi-token* decoding, on popular mathematical reasoning, GSM8K (Cobbe et al., 2021a) and MATH500 (Luo et al., 2024), and coding benchmarks, HUMANEVAL (Chen et al., 2021) as well as Sudoku (Shahab, 2025):

1. **Does any-order decoding help for text?** Popular sampling approaches for MDLMs select positions to unmask based on confidence (e.g. token probability (Chang et al., 2022) or entropy (Kim et al., 2025; Ye et al., 2025b)). We find that these any-order decoding algorithms either sample a large portion of tokens in a left-to-right order or underperform left-to-right sampling. For example, on GSM8K, the best configuration of any-order entropy decoding samples $\sim 50\%$ of tokens left-to-right. Without block sizes (Arriola et al., 2025) that enforce a semi-auto-regressive (AR) left-to-right structure, any-order significantly affects performance, see fig. 2. A notable exception where any-order sampling provides a significant benefit is Sudoku. We include additional analysis in section B and in table 7 in section B.

2. **Does fixed parallel token decoding work?** We observe that even decoding two tokens in parallel at a time can severely hurt model performance across all tasks, see table 1. The resulting distributions also have high KL with respect to a one-token sampling algorithm, see table 6.

| Parallel Tokens | GSM8K | | MATH500 | | HumanEval | | Sudoku | |
| --- | --- | --- | --- | --- | --- | --- | --- | --- |
| | LLaDA | Dream | LLaDA | Dream | LLaDA | Dream | LLaDA | Dream |
| 1 | **76.95** | **75.73** | **33.4** | **29.6** | **16.46** | **51.82** | 47.64 | **61.26** |
| 2 | 62.31 | 57.69 | 19.6 | 16.6 | 4.87 | 20.12 | **50.79** | 57.59 |
| 4 | 33.58 | 28.50 | 7.0 | 3.6 | 4.87 | 12.19 | 29.32 | 42.93 |

**Table 1:** MDLMs **can generate multiple fixed tokens in parallel, but this degrades accuracy.** We decode $1, 2, 4$ tokens in parallel, with block any-order (8) entropy decoding. We note that decoding even two tokens in parallel leads to a significant drop on all tasks but Sudoku.

These findings show that the decoding order from NTP models is performant for MDLM models, despite their any-order and multi-token decoding capabilities. Despite these findings, we show that the additional capacities offered by MDLMs have many possible benefits.

## 4 RETHINKING REASONING AND SAMPLING WITH MDLMs

MDLMs are trained to in-fill sequences by modeling the distributions $p_\theta(x^j \mid \mathbf{x}_{\text{UNMASKED}}, \mathbf{c})$ for masked positions $j \in \text{MASK-SET}$ given unmasked text $\mathbf{x}_{\text{UNMASKED}}$ and a context $\mathbf{c}$. Typically, MDLMs are prompted similarly to NTP models, and the distributions of the masked positions are used only for sampling a small *fixed* number of positions. The remaining distributions are discarded. In this work, we show that the ability of MDLMs to in-fill and to access the distribution of *all* masked positions unlocks many new sampling and post-training capabilities.

- **Reasoning-as-Infilling for Control, Early Exits, and Post-Training Benefits**. We propose pre-filling a user-specified prompt in multiple parts of the sequence. Specifically for reasoning tasks, we first pre-fill a reasoning template that differentiates between reasoning and answer positions, then infill with the MDLM model. This method of prompting enables controlling the length of the reasoning process, and measuring the uncertainty of the answer block during the reasoning process for early exiting. We also demonstrate how this approach supports new post-training directions for MDLMs.

- **Multi-token Entropy Decoding**. We introduce MED, an adaptive multi-token decoding algorithm that controls the error incurred by multi-token decoding by decoding multiple positions only if the conditional entropies of the decoded positions fall below a threshold.

**Assumptions.** We assume that the masked conditional distributions learned by the MDLM model define a consistent joint distribution (Majid et al., 2025).

### 4.1 REASONING-AS-INFILLING WITH MDLMs

Generally, NTP models are controlled at inference-time with a prompt prefix that is inserted at the beginning of the sequence. However, for MDLMs we propose pre-filling the output sequence with user-specified tokens. In the case of reasoning tasks, where a model produces a reasoning trace prior to answering, we can pre-fill the output sequence with a reasoning template that distinguishes the reasoning and answer token positions:

$$\left[ \underbrace{[\text{MASK}]_1 \quad [\text{MASK}]_2 \quad \ldots \quad [\text{MASK}]_k}_{\text{reasoning block}} \quad \text{<Answer Delimiter>} \quad \underbrace{[\text{MASK}]_{k+1} \quad \ldots \quad [\text{MASK}]_L}_{\text{answer block}} \right]$$

Here the answer delimiter is a user-specified choice (e.g. *"The answer is: "* for math tasks, or function definitions for a coding task). In this reformulation of prompting, the context $\mathbf{c}$ now includes both the prompt and the answer delimiter, see fig. 1. By distinguishing between reasoning and answer positions, reasoning-as-infilling offers several advantages for sampling and post-training.

**Early exits.** By designating explicit answer block positions, reasoning-as-infilling enables measuring answer uncertainty *while generating the reasoning trace*. A measure of uncertainty is the entropy of the answer block given the unmasked reasoning positions. This joint entropy requires additional estimation as MDLMs only provide access to the marginals $p_\theta(a^i \mid \mathbf{r}_{\text{UNMASKED}}, \mathbf{c})$. However,

we show that the marginal distributions can be used to upper-bound the joint entropy,

$$H_{\text{UB}} := \sum_{j \in \text{ANSWER-BLOCK}} H(a_j \mid \mathbf{r}_{\text{UNMASKED}}, \mathbf{c}) \ \geq \ H(\mathbf{a} \mid \mathbf{r}_{\text{UNMASKED}}, \mathbf{c}) \tag{3}$$

See section G for a proof. Using this quantity, we propose **early exiting based on the answer uncertainty upper-bound** $H_{\text{UB}}$. That is, given a partial reasoning trace, $\mathbf{r}_{\text{UNMASKED}}$, we skip filling in the remaining reasoning tokens if the answer-entropy upper bound falls below a user-specified threshold $\gamma$, $H_{\text{UB}} < \gamma$.

**Post-training MDLMs with reasoning-as-infilling.** Typically, post-training a model to reason uses expensive human demonstrations (Ouyang et al., 2022). Alternatively, Zelikman et al. (2022); Phan et al. (2023); Ruan et al. (2025) have demonstrated that post-training on model generated reasoning traces provides an alternative for improving performance (Zelikman et al., 2022; Phan et al., 2023; Ruan et al., 2025). These methods work off the principle that sampling reasoning traces from the posterior $p_\theta(\mathbf{r} \mid \mathbf{c}, \mathbf{a})$ and then training on these sample can increase the likelihood of generating correct answers. However, sampling from the posterior $p_\theta(\mathbf{r} \mid \mathbf{a}, \mathbf{c})$ is intractable for NTP models, therefore, Phan et al. (2023); Zelikman et al. (2022) make use of approximate sampling methods, which require either significant prompt engineering or training another model to yield reasoning traces given answer hints.

With reasoning-as-infilling in MDLMs, one can *simply* pre-fill the answer block positions to enable sampling from the posterior distribution, without prompt engineering or having to train another model. These posterior traces can be used for post-training in several ways, including with STaR (Zelikman et al., 2022), maximum marginal log-likelihood training (Phan et al., 2023; Murphy, 2023), or maximizing the likelihood on the answer and the posterior reasoning traces: $\max_\theta \sum_{i=1}^N \log p_\theta(\mathbf{a}_i, \mathbf{r}_i \mid \mathbf{c}_i)$ where the posterior reasoning traces are generated by the model, $\mathbf{r}_i \sim p_\theta(\mathbf{r}_i \mid \mathbf{c}_i, \mathbf{a}_i)$.

**Scoring partial reasoning traces when post-training.** Existing fine-tuning algorithms, such as GRPO (Shao et al., 2024) and RLOO (Ahmadian et al., 2024), do not make use of posterior samples but score the generations upon completion. These algorithms can benefit from intermediate rewards (Silver et al., 2016). Recent work shows that guiding the generation process with intermediate rewards produces samples that improve model fine-tuning (Zhang et al., 2024). These intermediate rewards are generally provided by an *external* pre-trained process reward model (PRM) (Lightman et al., 2023; Zhang et al., 2024; 2025b). Reasoning-with-infilling, given the answer, allows MDLMs to score arbitrary reasoning traces at intermediate steps. Given a partial reasoning trace $\mathbf{r}_{\text{UNMASKED}}$ and an answer $\mathbf{a}^*$, we can score $\mathbf{r}_{\text{UNMASKED}}$ with:

$$\phi(\mathbf{r}_{\text{UNMASKED}} \mid \mathbf{c}, \mathbf{a}^*) := \sum_{j \in \text{ANSWER-BLOCK}} \log p_\theta(a_j = a_j^* \mid \mathbf{c}, \mathbf{r}_{\text{UNMASKED}}). \tag{4}$$

The intuition behind the equation is that when the likelihood of individual answer tokens is higher for the reasoning trace $\mathbf{r}_{\text{UNMASKED}}$, then $\mathbf{r}_{\text{UNMASKED}}$ is often more likely to produce the answer.

### 4.2 Multi-token Entropy Decoding

As MDLMs learn the conditional distribution $p_\theta(x^j \mid \mathbf{x}_{\text{UNMASKED}})$ for all masked tokens, they support unmasking multiple tokens in parallel. However, decoding even two positions, $x^i$ and $x^j$ in parallel can result in samples that may not be likely under the MDLM joint distribution $p_\theta(\mathbf{x})$, as typically $p_\theta(x^i, x^j \mid \mathbf{x}_{\text{UNMASKED}}) \neq p_\theta(x^i \mid \mathbf{x}_{\text{UNMASKED}}) p_\theta(x^j \mid \mathbf{x}_{\text{UNMASKED}})$. In table 1, we observe that decoding even 2 tokens in parallel hurts task performance.

However, for any set of positions $A \subseteq \text{MASK-SET} \subseteq \{1, \dots, L\}$, we can upper bound the Kullback-Leibler (KL) divergence between the joint distribution $p_\theta(\mathbf{x}^A \mid \mathbf{x}_{\text{UNMASKED}}, \mathbf{c})$ and the factorized distribution $\prod_{i \in A} p_\theta(x^i \mid \mathbf{x}_{\text{UNMASKED}}, \mathbf{c})$ with the sum of the entropies of the masked tokens:

$$\text{KL}\left( p_\theta(\mathbf{x}^A \mid \mathbf{x}_{\text{UNMASKED}}, \mathbf{c}) \middle\| \prod_{i \in A} p_\theta(x^i \mid \mathbf{x}_{\text{UNMASKED}}, \mathbf{c}) \right) \leq \sum_{i \in A} H(x^i \mid \mathbf{x}_{\text{UNMASKED}}, \mathbf{c}) \tag{5}$$

where $H$ is the entropy of the distribution $p_\theta(x^i \mid \mathbf{x}_{\text{UNMASKED}}, \mathbf{c})$. For a proof, see section G.

In this work, we propose multi-token entropy decoding, which makes use of the entropies of the masked positions $x^j$ to decide whether to decode multiple positions in parallel. Given unmasked

text $\mathbf{x}_{\text{UNMASKED}}$, a decoding threshold $\lambda$ and a maximum number of tokens to be decoded $k_{\max}$, we propose the following definition of the set $A$ for selecting positions to un-mask. In MED, we sort the position entropies in an ascending order and decode positions that satisfy $H(x^i \mid \mathbf{x}_{\text{UNMASKED}}, \mathbf{c}) < \lambda$ and select $k_{\max}$ such tokens. If no position has entropy lower than $\lambda$, we choose the position with the lowest entropy.

MED allows for upper bounding the Kullback-Leibler divergence in eq. (5) by $\lambda k_{\max}$, controlling the error incurred by multi-token decoding.

## 5 EXPERIMENTS

In this section, we examine the inference-time and post-training benefits of reasoning-as-infilling, such as (1) early-exits based on answer certainty $H_{\text{UB}}(\mathbf{a} \mid \mathbf{r}_{\text{UN-MASKED}}, \mathbf{c})$ and (2) the ability to bootstrap and score reasoning traces given (question, answer) pairs. In section A, we study the effectiveness of multi-token entropy decoding (MED) for parallel sampling.

### 5.1 THE BENEFITS OF REASONING-AS-INFILLING

**Early exits.** We investigate the inference-time benefits of reasoning-as-infilling on two mathematical reasoning datasets, GSM8k (Lightman et al., 2023) and MATH500 (Cobbe et al., 2021b), with the Dream 7B and LLaDA-8B models.

For both tasks, we consider a generation length $L = 256$ with block size 32. We pre-fill the answer delimiter *"The answer is \boxed{.."*, and allocate 10 answer tokens. As a baseline, we compare against allocating a sequence of length 256 with no-reasoning template. For sampling, we examine early exits with one-token decoding and MED with $\lambda = 0.2$.

In table 2, we observe:

- For both Dream and LLaDA, early exiting reduces the total number of NFEs, and increasing the early exit threshold $\gamma$ enables trading faster inference for task accuracy. For example, for LLaDA, we observe a 23% speed up on one-token entropy decoding with only a $< 1\%$ drop in performance versus baseline reasoning template. Early exits combined with MED provides further savings. LLaDa with MED and $\gamma = 0.1$ outperforms the base configuration on GSM8k with a $3.3\times$ speedup. In section F, we provide examples of reasoning traces with varying exit thresholds.

- Notably, the benefits of early exits are more pronounced for LLaDA than Dream, which requires higher exit thresholds for speed-ups. This may be due to Dream's adaption from an NTP model (Gong et al., 2024; Ye et al., 2025a). See section B for a discussion of the sampling behavior of Dream and LLaDA.

| Model | Sampler | GSM8K | | | Math500 | | |
|-------|---------|-------|------|------|---------|------|------|
| | | **Exit Param** | **NFEs ↓** | **Acc.** | **Exit Param** | **NFEs ↓** | **Acc.** |
| LLaDA | $\sigma_{\text{ENTROPY}, k=1}$ | NO TEMPLATE | 256 | 76.6 | NO TEMPLATE | 256 | **33.8** |
| LLaDA | $\sigma_{\text{ENTROPY}, k=1}$ | NO EXIT | 256 | 79.4 | NO EXIT | 256 | 33.4 |
| LLaDA | $\sigma_{\text{ENTROPY}, k=1}$ | $\gamma = 0.1$ | 193 | 78.6 | $\gamma = 0.3$ | 221 | 31.9 |
| LLaDA | $\sigma_{\text{MED}, \lambda=0.2}$ | NO EXIT | 94 | **79.9** | NO EXIT | 143 | 33.4 |
| LLaDA | $\sigma_{\text{MED}, \lambda=0.2}$ | $\gamma = 0.1$ | **77** | 79.3 | $\gamma = 0.3$ | 129 | 32.0 |
| Dream | $\sigma_{\text{ENTROPY}, k=1}$ | NO TEMPLATE | 256 | **80.1** | NO TEMPLATE | 256 | 33.4 |
| Dream | $\sigma_{\text{ENTROPY}, k=1}$ | NO EXIT | 256 | 79.8 | NO EXIT | 256 | **35.6** |
| Dream | $\sigma_{\text{ENTROPY}, k=1}$ | $\gamma = 0.7$ | 225 | 76.7 | $\gamma = 0.7$ | 245 | 33.2 |
| Dream | $\sigma_{\text{MED}, \lambda=0.2}$ | NO EXIT | 135 | 79.2 | NO EXIT | 147 | 35.6 |
| Dream | $\sigma_{\text{MED}, \lambda=0.2}$ | $\gamma = 0.7$ | **121** | 79.3 | $\gamma = 0.7$ | 141 | 35.4 |

**Table 2: Early-exits can accelerate MDLM inference.** We evaluate reasoning-as-infilling with early exits on a generation length of 256. Varying the early exit threshold $\gamma$ enables trading faster inference for task accuracy. Lower values of $\gamma$ preserve performance.

Next, we investigate how a dataset of question-answer pairs $\{(\mathbf{c}_i, \mathbf{a}_i)\}_{i=1}^{N}$, can be used to analyze and improve MDLMs.

| Model | Posterior Reasoning Scores | |
|---|---|---|
| | Qwen-PRM | GPT-4o |
| LLaDA Base | 0.31 | 0.36 |
| LLaDA Instruct | 0.38 | 0.43 |

Table 3: **The MDLM reasoning posterior yields high-quality traces for problems that the original instruct-tuned model fails to solve.** We perform posterior inference on the 1419 training samples that LLaDa-8B Instruct with greedy decoding fails to solve, and evaluate the resulting traces with two judges, QWEN2.5-MATH-PRM (Zhang et al., 2025a) and GPT-4o (Hurst et al., 2024). Both judges rate $\sim 40\%$ of the instruct-tuned reasoning chains as correct. Notably, even the posterior reasoning chains from the base model are rated as $> 31\%$ correct.

| Model | Posterior Reasoning Scores | |
|---|---|---|
| | Qwen-PRM | GPT-4o |
| Llama *STaR* ($L = 256$) | 0.39 | 0.33 |
| Llama *STaR* ($L = 512$) | 0.39 | 0.40 |
| LLaDA Base ($L = 128$) | 0.34 | 0.41 |
| LLaDA Instruct ($L = 128$) | 0.46 | 0.52 |

Table 4: **The MDLM reasoning posterior scores higher than Llama3-8B STaR reasoning traces.** We perform posterior inference on the 265 test samples that Llama3-8B Instruct with greedy decoding fails to solve, and evaluate the traces similar to table 3. We observe that the LLaDA instruct model posterior traces score higher than the Llama STaR reasoning traces generated when the answer is provided as a hint.

**The answer posterior is a source of high-quality reasoning traces.** Here, we evaluate reasoning traces $\mathbf{r}$ generated from the posterior distribution $p_\theta(\mathbf{r} \mid \mathbf{c}, \mathbf{a})$. A key challenge for training better reasoning models is collecting high quality reasoning traces (Zelikman et al., 2022). We investigate whether the MDLM posterior distribution can provide these traces, even when an MDLM incorrectly solves the original task. To do this, we utilize question-answer pairs from GSM8k (Lightman et al., 2023). In this experiment, we generate samples from the LLaDA-8B Instruct model with MED and any-order decoding. On the GSM8k training dataset, the model answers 1419 out of 7473 problems incorrectly. We use these 1419 question-answer pairs to generate reasoning traces from the MDLM posterior (i.e. with the answer pre-filled). We also generate reasoning traces with the base model (without instruction tuning).

To evaluate these reasoning traces for correctness, we use GPT4o (Hurst et al., 2024), and the Qwen2.5-Math-7B PRM (Zhang et al., 2025a), see section C for the system instructions to the GPT4o model. In table 3 observe that both judge models rate $\sim 40\%$ of the posterior reasoning traces as correct. In section D, we include examples of reasoning traces generated from the posterior and the regular model with different judge labels. We observe that the posterior traces judged correct by GPT4o contain accurate reasoning steps, correcting the original model's behavior. Notably, we also observe that reasoning-as-infilling elicits correct reasoning chains ($> 30\%$) from the base model checkpoint on these problems. In table 4, we analyze the reasoning traces for problems that Llama3-8B Instruct (Dubey et al., 2024) solves incorrectly. We note that the LLaDA-8B base and instruct models generate posterior reasoning traces that score higher than the LLama model prompted with the answer as a hint.

**Posterior data can be used to improve base models.** Here, we examine the effectiveness of post-training on the post-hoc reasoning dataset generated on the full GSM8k training set with the base model. We post-train the LLaDA-8B Base model using LoRA (Hu et al., 2022).

In table 5, we observe that fine-tuning the model on the posterior generated base model data significantly improves performance ($+14.9\%$). As a benchmark, we observe that fine-tuning on the GSM8k *human annotated* reasoning traces produces similar results. These results provide evidence that maximizing the log-likelihood $\log p_\theta(\mathbf{a}, \mathbf{r} \mid \mathbf{c})$ on the posterior reasoning traces improves accuracy on reasoning tasks. We include additional training details section H.

| Model | Post-training Data | GSM8K Test Acc. |
|---|---|---|
| LLaDA 8B-Base (No template) | - | 13.9 % |
| LLaDA 8B-Base (With template) | - | 51.2 % |
| Finetuned* LLaDA 8B-Base | GSM8k $(\mathbf{c}, \mathbf{r}_{\text{gold}}, \mathbf{a}_{\text{gold}})$ $(n = 7473)$ | 64.6 (+13.4) % |
| Finetuned* LLaDA 8B-Base | GSM8k Posterior $(\mathbf{c}, \mathbf{r}_{\text{posterior}}, \mathbf{a}_{\text{gold}})$ $(n = 7473)$ | 66.1 (+14.9) % |
| LLaDA-8B Instruct | Misc. Instruction Data $(n = 4.5 \text{ million})$ | 75.96 % |

**Table 5: Fine-tuning the base model with posterior-generated data improves performance.** Fine-tuning on GSM8k training data and posterior reasoning traces boosts accuracy to $64.6\%$ and $66.1\%$, respectively. Finetuned* indicates LoRA (Hu et al., 2022) fine-tuning.

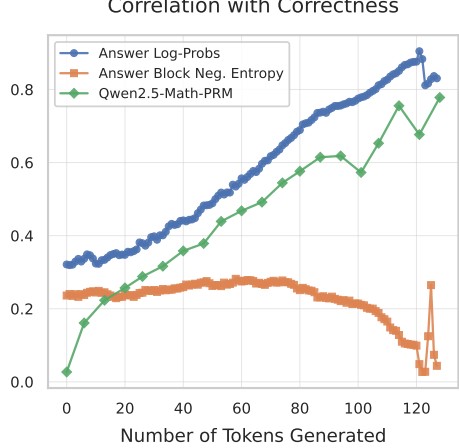

**Figure 3: MDLMs enable scoring their own reasoning process without an external process verifier.** We score GSM8k reasoning traces, generated left-to-right with LLaDA, at intermediate steps using (a) gold answer log probabilities, (b) the answer block entropy bound, and (c) an 8B process reward model (PRM) (Zhang et al., 2025b). Gold answer probability at intermediate steps is more predictive of final correctness than PRM scores. Even without gold labels, the answer block entropy is weakly correlated with correctness.

**Scoring partial reasoning traces without an external model.** Zhang et al. (2024) show that using intermediate process rewards for sampling can improve model fine-tuning. However, training these process rewards requires training an external model. In fig. 3, we compare various strategies for estimating the final correctness of partial reasoning traces given intermediate rewards. For further details see section E

Using the LLaDA-8B Instruct model, we greedily sample solutions on the GSM8k test set, left-to-right, 1 token at a time. We then compute the Pearson correlation between intermediate rewards and the correctness of the final output. During reasoning, the intermediate reasoning process defined using the answer log probabilities $\log p_\theta(a_j = a_j^* \mid \mathbf{c}, \mathbf{r}_{\text{UN-MASKED}})$ are more strongly correlated with final answer correctness at intermediate steps than a pretrained 7B parameter process reward model (Zhang et al., 2025a).

Our results provide evidence that MDLM pre-training offers other new post-training capabilities: low-quality reasoning chains could be terminated early or filtered, the reasoning process could be steered towards correct solutions, reflection tokens could be automatically inserted at reasoning failures, and new sources of dense feedback could incorporated into fine-tuning objectives.

## 6 DISCUSSION AND LIMITATIONS

Much of the current tooling around pre-training, post-training, and inference for text generation has been built around a key modeling choice: next-token prediction training. MDLMs are an expressive class of models trained to in-fill masked sequences, requiring additional training and inference compute. In our work, we find that this additional compute has many uses beyond accelerating inference and warrants *rethinking* how these models are utilized. For instance, the ability to in-fill unlocks new prompting techniques, like the proposed reasoning-as-filling framework, along with new data generation and post-training methods.

Reasoning-as-infilling requires specifying the length of the reasoning and answer blocks, however, we note that the methods developed with reasoning-as-infilling can be paired with variable sequence length models (Wu et al., 2025b) that obviate the need to pre-specify the length of reasoning and answer tokens, and can enable dynamically scaling compute for harder problems.

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

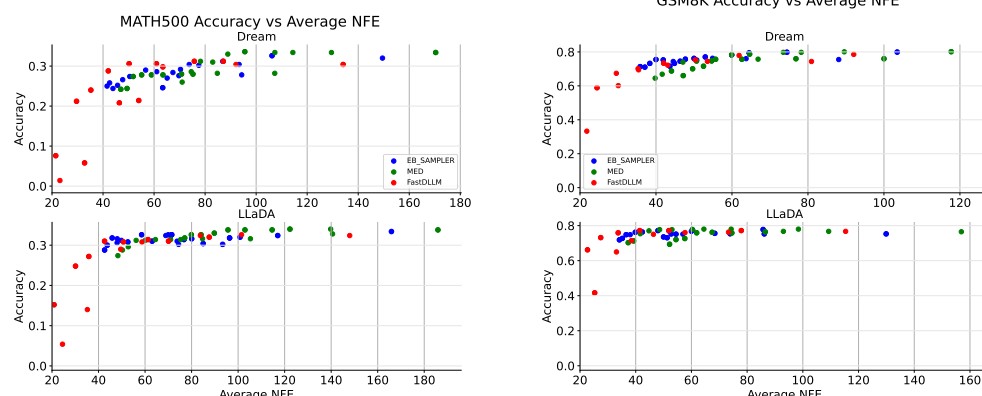

**Figure 4: Adaptive decoding enables parallel token decoding without any loss in performance.** In contrast to fixed token decoding, adaptive parallel decoding algorithms enable preserving performance while reduce the number of function evaluations. Here, we plot results from MED alongside two recently proposed adaptive samplers, EB-Sampler (Ben-Hamu et al., 2025) and Block-wise confidence-aware parallel decoding (Wu et al., 2025a). In this experiment, we generate sequences of length $L \in \{128, 256\}$ with varying thresholds for the three samplers. The best sampler depends on task and NFE budget. For instance, for a $2\times$ reduction in NFEs, MED yields the highest accuracies on MATH500, while producing comparable numbers to EB-sampler for GSM8k.

# A  ACCELERATED SAMPLING WITH MULTI-TOKEN ENTROPY DECODING

In this experiment, we show that entropy-thresholded multi-token decoding (MED) enables parallel decoding without incurring the distributional error and performance degradation that fixed multi-token decoding incurs.

For these experiments, we use two open-source MDLMs models, Dream-7B Instruct (Ye et al., 2025a) and LLaDA-8B Instruct (Nie et al., 2025). We consider two popular benchmarks, (1) GSM8k (Lightman et al., 2023), a mathematical reasoning dataset, and (2) HumanEval (Chen et al., 2021), a coding benchmark. As baselines, we consider the entropy decoding scheme $\sigma_{\text{ENTROPY},k}$ (Chang et al., 2022; Ye et al., 2025b), which decodes a fixed number of $k$ tokens in each step. We consider $k \in \{1, 2\}$.

For evaluations, we measure the task accuracy and the number of function evaluations (NFEs) for varying values of $k$ in the fixed token decoding scheme as well as varying values of $\lambda \in \{0.1, 0.2, 0.3\}$ in MED with $k_{\max} = 32$ as the maximum number of tokens decoded in parallel. We fix a generation length $L \in \{128, 256\}$ and a block size of 32. As baselines, we compare against EB-Sampler (Ben-Hamu et al., 2025) and block-wise confidence-aware parallel decoding (Wu et al., 2025a). In fig. 4, we compare the three adaptive decoding schemes with varying thresholds on the GSM8K, MATH500 datasets. We observe that the best sampler depends on task and NFE budget. For instance, for a $2\times$ reduction in NFEs, MED yields the highest accuracies on MATH500, while producing comparable numbers to EB-sampler for GSM8k.

In table 6, we observe:

- Decoding just $k = 2$ tokens in parallel results in a large drop in accuracy on GSM8k for both LLaDA and Dream ($> 40\%$). We observe that decoding $k = 2$ also leads to a significant increase in KL.

- MED with $\lambda = 0.2$, provides significant speed-ups and *no loss in accuracy* for both LLaDA and Dream. For HUMANEVAL, MED results in identical accuracy with a $2.2\times$ speed-up, and on GSM8k, we observe a $1.5\times$ speed-up with no loss in performance.

| Dataset | Sampler | Model | Accuracy ↑ | KL ↓ | NFEs ↓ |
|---|---|---|---|---|---|
| GSM8K | $\sigma_{\text{ENTROPY}, k=1}$ | LLaDa | **78.01** | 0.0 | 128.0 |
| | | Dream | **75.81** | 0.0 | 128.0 |
| | $\sigma_{\text{ENTROPY}, k=2}$ | LLaDa | 36.24 | 92.6 | 64.0 |
| | | Dream | 19.79 | 99.5 | 64.0 |
| | $\sigma_{\text{MED}, \lambda=0.1}$ | LLaDa | 78.01 | 0.5 | 88.4 |
| | | Dream | 75.81 | 0.4 | 92.5 |
| | $\sigma_{\text{MED}, \lambda=0.2}$ | LLaDa | 78.01 | 1.5 | 84.8 |
| | | Dream | 75.82 | 1.9 | 79.9 |
| | $\sigma_{\text{MED}, \lambda=0.3}$ | LLaDa | 77.86 | 2.7 | 81.2 |
| | | Dream | 75.44 | 3.3 | 75.7 |
| HumanEval | $\sigma_{\text{ENTROPY}, k=1}$ | LLaDa | 15.24 | 0.0 | 128.0 |
| | | Dream | **48.17** | 0.0 | 128.0 |
| | $\sigma_{\text{ENTROPY}, k=2}$ | LLaDa | 4.87 | 85.5 | 64.0 |
| | | Dream | 20.12 | 77.0 | 64.0 |
| | $\sigma_{\text{MED}, \lambda=0.1}$ | LLaDa | 15.24 | 0.7 | 70.0 |
| | | Dream | 48.17 | 0.5 | 68.5 |
| | $\sigma_{\text{MED}, \lambda=0.2}$ | LLaDa | 15.85 | 2.0 | 61.8 |
| | | Dream | 48.17 | 1.5 | 60.4 |
| | $\sigma_{\text{MED}, \lambda=0.3}$ | LLaDa | **16.46** | 3.6 | 57.8 |
| | | Dream | **48.17** | 2.2 | 57.0 |

**Table 6: MED enables parallel token decoding without any loss in performance.** We compare MED decoding with different $\lambda$ thresholds to entropy decoding with a fixed number of tokens $k \in \{1, 2\}$. We observe that MED significantly reduces the number of NFEs while matching accuracy and maintaining a low KL.

| Decoding order | GSM8K | | Math500 | | HumanEval | | Sudoku (4) | |
|---|---|---|---|---|---|---|---|---|
| | Llada | Dream | Llada | Dream | Llada | Dream | Llada | Dream |
| Left-to-right | 75.96 | 74.98 | 29.4 | 29.2 | 15.24 | **53.65** | 36.13 | 17.28 |
| Any-order decoding | 53.44 | 34.11 | 12.0 | 20.6 | 10.97 | 32.92 | **47.64** | **61.26** |
| Block any-order (8) | 76.95 | 75.73 | **33.4** | **29.6** | **16.46** | 51.82 | 38.74 | 44.50 |
| Block any-order (32) | **78.01** | **75.81** | 32.4 | 28.2 | 15.24 | 48.17 | **47.64*** | **61.26*** |
| Block any-order (64) | 74.14 | 57.69 | 30.0 | 28.2 | 14.63 | 47.56 | - | - |
| Llama3.1-8B | 70.81 | | 26.80 | | 62.20 | | 2.09 | |

**Table 7: Left-to-right sampling is a competitive sampling algorithm for reasoning and coding.** When performing entropy decoding (Ye et al., 2025b), we observe that full any-order sampling results in poor performance on all tasks but Sudoku. Left-to-right block decoding is required to make any-order sampling performant, and left-to-right sampling (block size = 1) is always within a few percent of the best configuration. We also observe in section B that performant block any-order configurations sample a large portion of tokens left-to-right.*For Sudoku, we consider sequences of length 32, otherwise we use a sequence length of 128.

# B MDLM ANY-ORDER SAMPLING BEHAVIOR

We study the effects of greedy any-order entropy decoding (Ye et al., 2025a) for LLaDA (Nie et al., 2025) and Dream (Ye et al., 2025a), as well as any-order with different **block lengths** (Sahoo et al., 2024; Arriola et al., 2025; Nie et al., 2025; Ye et al., 2025a). The block length is the contiguous region of consecutive positions considered by the sampling algorithm, where the model can decode in any order. Blocks are unmasked left-to-right.

We included our results in table 7. On Sudoku, any-order sampling significantly improves performance.[1] However, for the remaining datasets, left-to-right sampling with a block length of 1 is a competitive approach. In some cases (e.g. for Dream on HUMANEVAL (Chen et al., 2021)), left-to-right block length 1 sampling is the most performant configuration. Additionally, purely any-order decoding (i.e. when the block size = generation length), leads to a massive drop in performance.

**In what order are tokens decoded?** In table 8 and section B, we analyze the behavior of these different configurations on a portion of GSM8K and HUMANEVAL. We compute the fraction of non-EOS tokens decoded from the leftmost masked position, the average distance from the leftmost position, and the total number of non-EOS tokens. For GSM8K, we also include the average step at which the answer appears in the decoded sequence.

Notably, top performing block-length configurations often behave very autoregressively. On GSM8K, when the block size is 32, both LLaDA and Dream sample the leftmost unmasked position approximately 50% of the time. Additionally, the average distance of the unmasked position from the left-most mask is approximately 3 tokens. Gong et al. (2025) similarly observe the left-to-right sampling behavior of Dream for coding.

**Why does block sampling improve performance?** We find that that purely any-order decoding from current MDLMs results in less auto-regressive generation, fewer non-eos tokens, and very early answers, not utilizing the full allocated generation length. Reviewing samples from any-order decoding, we observe two specific pathological behaviors: 1) Models first greedily decoding low entropy *end-of-text* tokens, leading to shorter or empty texts that do not fully utilize the assigned tokens, and 2) *decoding only an answer, or decoding answers first, before reasoning chains*.

| Config | Model | Acc. | % Leftmost | Dist. Left | Non-EOS Tokens | Answer Step |
|--------|-------|------|-----------|-----------|----------------|-------------|
| Block(1) | Dream | 76.4% | 100.0% | 0.0 | 105.1 | 78.1 |
| | LLaDA | 79.0% | 100.0% | 0.0 | 115.3 | 84.3 |
| Block(32) | Dream | 77.6% | 52.1% | 2.9 | 103.1 | 76.3 |
| | LLaDA | 76.8% | 47.1% | 3.3 | 112.3 | 82.8 |
| AO(128) | Dream | 34.2% | 73.1% | 6.5 | 24.3 | 18.7 |
| | LLaDA | 53.4% | 40.8% | 20.2 | 75.0 | 16.9 |

**Table 8: Decoding behavior, GSM8K** We evaluate the autoregressiveness of different sampling configurations by measuring the percent of non-EOS tokens decoded from the leftmost position, the average distance of these positions from left, the total number of non-EOS tokens, and at what timestep the answer is decoded. We consider generation lengths of 128 on a portion of GSM8K ($n = 500$)

| Config | Model | Acc. | % Leftmost | Dist. Left | Non-EOS Tokens |
|--------|-------|------|-----------|-----------|----------------|
| Block(1) | Dream | 53.7% | 100.0% | 0.0 | 95.0 |
| | LLaDA | 11.0% | 100.0% | 0.0 | 119.8 |
| Block(32) | Dream | 48.2% | 43.1% | 3.8 | 96.8 |
| | LLaDA | 15.2% | 44.7% | 4.1 | 119.5 |
| AO(128) | Dream | 32.9% | 44.5% | 6.8 | 56.4 |
| | LLaDA | 15.2% | 29.7% | 19.7 | 123.8 |

**Table 9: Decoding behavior, HumanEval** Similar to table 8, we measure autoregressiveness for generation lengths of 128, on a portion of HumanEval ($n = 500$)

---

[1]Of note, on Sudoku, diffusion models with auto-regressive sampling significantly outperform Llama 8B. This may reflect benefits of the MDLM training objective.

## C  EVALUATING REASONING TRACE CORRECTNESS

We evaluate reasoning trace correctness with GPT4o (Hurst et al., 2024). We use the following template to evaluate the reasoning chain:

```
SYSTEM_INSTRUCTIONS = """
You are a strict grader.
You are given a question and a model's response.

Return a answer of either:
- 1 if both the model's final answer AND reasoning chain are correct.
- 0 otherwise.

Your final answer should be the digit 0 or 1.
"""

PROMPT_TEMPLATE = """Grade the submission.

[Question]
{question}

[Model Response]
{reasoning}

[Reference Answer]
{reference_answer}
"""
```

We use 512 tokens for all judge models, and default hyperparameters.

## D  REASONING TRACE EXAMPLES

Below, we include reasoning traces from the model $p_\theta(\mathbf{a} \mid \mathbf{r}, \mathbf{c})p_\theta(\mathbf{r} \mid \mathbf{c})$ and the posterior traces $p_\theta(\mathbf{r} \mid \mathbf{a}, \mathbf{c})$. We include samples where LLaDA originally fails to correctly answer the question. In two cases, the posterior succesfully provides an alternative correct reasoning trace, and in two it fails to do so.

### D.1  CORRECT POSTERIOR TRACES

> **Example 1**
>
> **Question:** Ms. Warren ran at 6 mph for 20 minutes. After the run, she walked at 2 mph for 30 minutes. How many miles did she run and walk in total?
>
> **Incorrect LLaDA output:**
>
> ```
> 1 Ms. Warren ran at 6 mph for 20 minutes, which is $\frac{20}{60} =
>     \frac{1}{3}$ hours. So, she ran $6 \times \frac{1}{3} = 2$
>     miles.
> 2
> 3 Ms. Warren walked at 2 mph for 30 minutes, which is $\frac{30}{60}
>     = \frac{1}{2}$ hours. So, she walked $2 \times \frac{1}{2} = 2
>     miles as well.
> 4
> 5 The answer is \boxed{4} miles.
> ```
>
> **Correct LLaDA posterior:**
>
> ```
> 1 Ms. Warren ran at 6 mph for 20 minutes, which is 20/60 = 1/3 of an
>     hour.
> 2 So, she ran 6 mph * 1/3 hour = 2 miles.
> ```

```
3 After the run, she walked at 2 mph for 30 minutes, which is 30/60 =
      1/2 of an hour.
4 So, she walked 2 mph * 1/2 hour = 1 mile.
5 In total, Ms. Warren ran and walked 2 miles + 1 mile = 3 miles.
6 The answer is \boxed{3}
```

### Example 2

**Question:** Each week Jaime saves $50. Every two weeks she spends $46 of her savings on a nice lunch with her mum. How long will it take her to save $135?

**Incorrect LLaDA output:**

```
1 Jaime saves $50 each week, so in two weeks she saves $50 x 2 = $100.
2 Every two weeks she spends $46, so in two weeks she saves $100 -
      $46 = $54.
3 To save $135, she needs to save $135 - $54 = $81 more.
4 Since she saves $54 every two weeks, it will take her $81 / $54 = 3
      more two weeks to save $135.
5 The answer is \boxed{3}+2 = 6 weeks.
```

**Correct LLaDA posterior:**

```
1 Jaime saves $50 each week, so in two weeks she saves $50 x 2 = $100.
2 However, she spends $46 every two weeks, so her net savings every
      two weeks is $100 - $46 = $54.
3 To find out how long it will take her to save $135, we divide $135
      by her net savings every two weeks: $135 / $54 = 2.5.
4 Therefore, it will take her 2.5 two weeks to save $135.
5 The answer is \boxed{5}
```

## D.2 INCORRECT POSTERIOR TRACES

### Example 1

**Question:** Tommy is making 12 loaves of bread. He needs 4 pounds of flour per loaf. A 10-pound bag of flour costs $10 and a 12-pound bag costs $13. When he is done making his bread, he has no use for flour and so he will throw away whatever is left. How much does he spend on flour if he buys the cheapest flour to get enough?

**Incorrect LLaDA output:**

```
1 Tommy needs 12 loaves of bread, and each loaf requires 4 pounds of
      flour.
2 Therefore, he needs a total of 12 * 4 = 48 pounds of flour.
3
4 The cheapest option is to buy 12-pound bags of flour.
5 Since Tommy needs 48 pounds, he will need 48/12 = 4 bags.
6 Each bag costs $13, so the total cost for 4 bags is 4 * 13 = $52.
7
8 The answer is \boxed{52}.
```

**Incorrect LLaDA posterior:**

```
1 Tommy needs 12 loaves of bread, so he needs a total of 12 * 4 = 48
      pounds of flour.
2 If he buys 10-pound bags, he will need 48/10 = 4 bags,
3 which will cost 4 * 10 = 40.
4 If he buys 12-pound bags, he will need 48/12 = 4 bags, which will
      cost 4 * 13 = $52.
```

```
5 Therefore, Tommy should buy 10-pound bags to save money.
6 The answer is \boxed{50}.
```

### Example 2

**Question:** Thomas started saving for a car almost 2 years ago. For the first year, his weekly allowance was $50. In the second year, he got a job that pays $9 an hour at a coffee shop and worked 30 hours a week, so his parents discontinued his allowance. If the car he wants to buy is $15,000 and he spends $35 a week on himself, how much more money does Thomas need to buy the car by the end of the 2 years?

**Incorrect LLaDA output:**

```
1 In the first year, Thomas earned $50 x 52 = $2600.
2 In the second year, Thomas earned $9 x 30 x 52 = 7980.
3 Thomas saved $2600 + $7980 = 10580.
4 Thomas spent $35 x 52 = 1820.
5 Thomas still needs $15000 - 10580 - 1820 = 2600 more.
6 The answer is \boxed{2600}.
```

**Incorrect LLaDA posterior:**

```
1 In the first year, Thomas earned $50 x 52 = $2600.
2 In the second year, Thomas earned $9 x 30 x 52 = 8760.
3 Thomas spent $35 x 52 x 52 = 8360.
4 In total, Thomas saved $2600 + $8760 - $8360 = 3000.
5 To buy the car, Thomas still needs $15000 - 3000 = $2000.
6 The answer is \boxed{2000}
```

## E   SCORING POSTERIOR REASONING CHAINS WITH MDLMS

Given that posterior sampling yields both high-quality and low-quality reasoning chains, a natural question is: *Can we filter these chains without an external model?*

We find that answer block log probabilities computed with reasoning-as-infilling can be used to filter these reasoning chains, and identify traces that GPT4-o rates as correct. To do this, we iteratively unmask each generated posterior chain, left-to-right, and average the answer log probabilities over all time-steps. For LLaDA, we observe that correct reasoning traces correspond to higher average scores, see fig. 5. We also find that thresholding these answer block entropy scores results in a reasonably performant classifier, with AUC=0.74, see fig. 6.

## F   EARLY EXIT SAMPLES

Below we demonstrate outputs from LLaDA on the GSM8k dataset. For a sample question from the test set, we generate a sequence of length 256 with the following settings"

- Sampling from the model, without using the reasoning template or early exits.

- Reasoning template with no exits.

- Reasoning template with varying exit thresholds.

- Reasoning template with the model producing the answer with no reasoning tokens decoded.

### Example 1

**Question:** Janet's ducks lay 16 eggs per day. She eats three for breakfast every morning and bakes muffins for her friends every day with four. She sells the remainder at the farmers' market daily for $2 per fresh duck egg. How much in dollars does she make every day at the

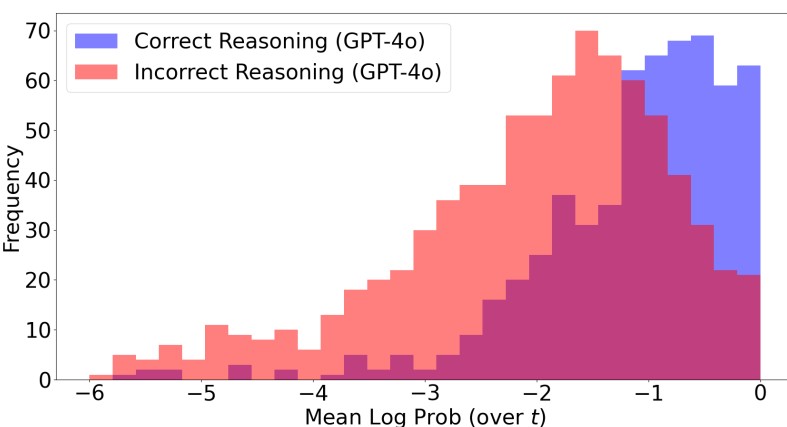

**Figure 5: Distribution of answer block log probability scores for posterior samples.** To score the 1419 posterior reasoning traces, we iteratively unmask each chain left-to-right, and compute the average answer block log probabilities across all timesteps. Posterior chains rated as correct by GPT4-o tend to have higher scores.

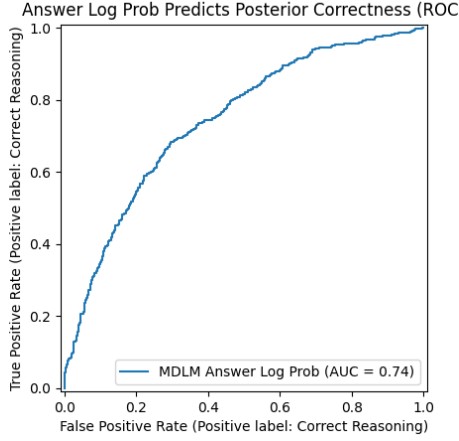

**Figure 6: Answer block log probability scores predict posterior reasoning trace quality.** The MDLM average answer block log probabilities can simply be thresholded to provide a classifier for predicting GPT4-o reasoning chain judgments. This classifier provides a potential method for filtering low-quality posterior samples (e.g. before fine-tuning).

---

farmers' market?

**LLaDA (No reasoning template, No Early Exits):**

```
1
2 To determine how much Janet makes every day at the farmers' market,
     we need to follow these steps:
3 1. Calculate the total number of eggs laid by Janet's ducks per day.
4 2. Subtract the number of eggs Janet eats and bakes for her friends
     from the total number of eggs laid.
5 3. Determine the number of eggs she sells at the farmers' market.
6 4. Calculate the total revenue from selling the eggs.
7 Step 1: Janet's ducks lay 16 eggs per day.
```

```
8 Step 2: Janet eats 3 eggs for breakfast every morning and bakes 4
       eggs for her friends every day. Therefore, she eats and bakes a
       total of \(3 + 4 = 7\) eggs per day.
9 Step 3: The number of eggs she sells at the farmers' market is \(16
       - 7 = 9\) eggs per day.
10 Step 4: Janet sells each egg for $2. Therefore, the total revenue
       from selling 9 eggs per day is \(9 \times 2 = 18\) dollars.
11 So, the amount Janet makes every day at the farmers' market is
       \(\boxed{18}\).
```

**LLaDA output (Reasoning template, No Early Exit):**

```
1
2
3 To determine how much Janet makes every day at the farmers' market,
       we need to follow these steps:
4 Step 1: Calculate the total number of eggs laid by Janet's ducks
       per day.
5 Janet's ducks lay 16 eggs per day.
6 Step 2: Calculate the number of eggs Janet eats for breakfast per
       day.
7 Janet eats 3 eggs for breakfast every day.
8 Step 3: Calculate the number of eggs Janet bakes for her friends
       per day.
9 Janet bakes 4 eggs for her friends every day.
10 Step 4: Calculate the number of eggs left after Janet eats some and
       bakes some.
11 Total eggs laid - Eggs eaten for breakfast - Eggs bakes for friends
       = Eggs left
12 16 - 3 - 4 = 9
13 So, Janet has 9 eggs left to sell.
14 Step 5: Calculate the revenue from selling the eggs at the farmers'
       market.
15 Janet sells each egg for $2.
16 Eggs left * Price per egg = Total revenue
17 9 * 2 = 18
18 Therefore, Janet makes $18 every day at the farmers' market.
19 The answer is \boxed{18}.
```

**LLaDA output (Reasoning template, Early Exit($\gamma = 0.5$)):**

```
1 To determine how much Janet makes every day at the farmers' market,
       we need to follow these steps:
2 Step 1: Calculate the total number of eggs laid by Janet's ducks
       per day.
3 <|mdm_mask|>...<|mdm_mask|> per<|mdm_mask|>...<|mdm_mask|> The
       answer is \boxed{18}.
```

**LLaDA output (Reasoning template, Early Exit($\gamma = 0.7$)):**

```
1 To determine<|mdm_mask|>...<|mdm_mask|> The answer is \boxed{18}.
```

**LLaDA output (Forced Answer):**

```
1 <|mdm_mask|>...<|mdm_mask|>. The answer is \boxed{14}.
```

# G  PROOFS

**MED KL upper-bound.**  Here we prove that for any set $A \subset \{1, \dots\}/\text{UN-MASKED}$, the following upper bound holds:

$$\text{KL}\left(p_\theta(\mathbf{x}^A \mid \mathbf{x}_{\text{UN-MASKED}}, \mathbf{c}) \middle\| \prod_{i \in A} p_\theta(x^i \mid \mathbf{x}_{\text{UN-MASKED}}, \mathbf{c})\right) \leq \sum_{i \in A} H(x^i \mid \mathbf{x}_{\text{UN-MASKED}}, \mathbf{c}) \quad (6)$$

Note that:

$$\text{KL}\left(p_\theta(\mathbf{x}^A \mid \mathbf{x}_{\text{UN-MASKED}}, \mathbf{c}) \middle\| \prod_{i \in A} p_\theta(x^i \mid \mathbf{x}_{\text{UN-MASKED}}, \mathbf{c})\right)$$

$$= \mathbf{E}_{p_\theta(\mathbf{x}^A \mid \mathbf{x}_{\text{UN-MASKED}}, \mathbf{c})}\left[\log p_\theta(\mathbf{x}^A \mid \mathbf{x}_{\text{UN-MASKED}}, \mathbf{c}) - \log \prod_{i \in A} p_\theta(x^i \mid \mathbf{x}_{\text{UN-MASKED}}, \mathbf{c})\right] \quad (7)$$

$$= \mathbf{E}_{p_\theta(\mathbf{x}^A \mid \mathbf{x}_{\text{UN-MASKED}}, \mathbf{c})}\left[\log p_\theta(\mathbf{x}^A \mid \mathbf{x}_{\text{UN-MASKED}}, \mathbf{c}) - \sum_{i \in A} \log p_\theta(x^i \mid \mathbf{x}_{\text{UN-MASKED}}, \mathbf{c})\right] \quad (8)$$

$$= -H(\mathbf{x}^A \mid \mathbf{x}_{\text{UN-MASKED}}, \mathbf{c}) + \sum_{i \in A} H(\mathbf{x}^i \mid \mathbf{x}_{\text{UN-MASKED}}, \mathbf{c}) \quad (9)$$

Now, since the entropy for discrete random variables is positive, $H(\mathbf{x}^A \mid \mathbf{x}_{\text{UN-MASKED}}, \mathbf{c}) \geq 0$, which implies:

$$-H(\mathbf{x}^A \mid \mathbf{x}_{\text{UN-MASKED}}, \mathbf{c}) + \sum_{i \in A} H(\mathbf{x}^i \mid \mathbf{x}_{\text{UN-MASKED}}, \mathbf{c}) \leq \sum_{i \in A} H(\mathbf{x}^i \mid \mathbf{x}_{\text{UN-MASKED}}, \mathbf{c}) \quad (10)$$

Hence, we have that for any set $A$, we have that:

$$\text{KL}\left(p_\theta(\mathbf{x}^A \mid \mathbf{x}_{\text{UN-MASKED}}, \mathbf{c}) \middle\| \prod_{i \in A} p_\theta(x^i \mid \mathbf{x}_{\text{UN-MASKED}}, \mathbf{c})\right) \leq \sum_{i \in A} H(x^i \mid \mathbf{x}_{\text{UN-MASKED}}, \mathbf{c}) \quad (11)$$

**Entropy upper-bound**  Next, we prove that:

$$H(\mathbf{a} \mid \mathbf{r}_{\text{UN-MASKED}}, \mathbf{c}) \leq H_{\text{UB}}(\mathbf{a} \mid \mathbf{r}_{\text{UN-MASKED}}, \mathbf{c}) \quad (12)$$

where $H_{\text{UB}}(\mathbf{a} \mid \mathbf{r}_{\text{UN-MASKED}}, \mathbf{c}) = \sum_i H(a^i \mid \mathbf{r}_{\text{UN-MASKED}}, \mathbf{c})$.

Next, we note that $\text{KL}(p_\theta(\mathbf{a} \mid \mathbf{r}_{\text{UN-MASKED}}, \mathbf{c}) \mid \prod p_\theta(a^i \mid \mathbf{r}_{\text{UN-MASKED}}, \mathbf{c}) \geq 0$, which implies that, similar to eq. (9), we have

$$-H(\mathbf{a} \mid \mathbf{r}_{\text{UN-MASKED}}, \mathbf{c}) + H_{\text{UB}}(\mathbf{a} \mid \mathbf{r}_{\text{UN-MASKED}}, \mathbf{c}) = \text{KL}(p_\theta(\mathbf{a} \mid \mathbf{r}_{\text{UN-MASKED}}, \mathbf{c}) \mid \prod p_\theta(a^i \mid \mathbf{r}_{\text{UN-MASKED}}, \mathbf{c})$$

$$-H(\mathbf{a} \mid \mathbf{r}_{\text{UN-MASKED}}, \mathbf{c}) + H_{\text{UB}}(\mathbf{a} \mid \mathbf{r}_{\text{UN-MASKED}}, \mathbf{c}) \geq 0 \quad (13)$$

$$H_{\text{UB}}(\mathbf{a} \mid \mathbf{r}_{\text{UN-MASKED}}, \mathbf{c}) \geq H(\mathbf{a} \mid \mathbf{r}_{\text{UN-MASKED}}, \mathbf{c}) \quad (14)$$

# H  FINE-TUNING DETAILS

We compare fine-tuning the LLaDA-8B Base (Nie et al., 2025) model on GSM8k (Cobbe et al., 2021b) reasoning data, versus posterior data sampled from the same model using the training questions and pre-filled answers.

**Data**  The posterior data is from LLaDA-8B-Base by pre-filling the correct answer and reasoning template, and sampling with entropy decoding and a block size of 128. For the gold GSM8k training data, we preprocess the data by removing the additional computations in angle brackets, and converting the "$\#\#\#\#$" format to our reasoning template. Additionally, unlike posterior data, GSM8k reasoning traces are of varying lengths. As a result, we truncate these traces to $L = 144$ tokens. When traces are longer, we truncate to the *last* 144 tokens. Approximately $\sim 12\%$ of GSM8k samples are truncated.

**Training**  We use a batch size of 1 per GPU, with 8 different noise levels per batch element. We use LoRA (Hu et al., 2022) with $r = 128, \alpha = 32$. We fine-tune the model using 2 Nvidia A100 GPUs, with a learning rate of $2.5 \times 10^{-6}$, and 32 gradient accumulation steps. We train both models for 3300 steps, or 30 epochs.

We modify the supervised fine-tuning code provided by Zhao et al. (2025).

**Sampling**  We greedily sample from both models with left-to-right with a block-size of 1. We allocate 144 tokens for both models, and do not directly pre-fill a reasoning template.

