# OpenReview forum: "Rethinking reasoning with Masked Diffusion Models"
_ICLR.cc/2026/Conference — Submitted to ICLR 2026_

### Official Review · Reviewer_s1QC · 2025-10-19

**Soundness:** 3
**Presentation:** 2
**Contribution:** 2
**Rating:** 2
**Confidence:** 4

**Summary:**

This paper investigates the parallel decoding behavior of MDLMs, revealing that even decoding two tokens per step leads to substantial performance degradation. To address this, the paper proposes an adaptive multi-token decoding algorithm, MED, which selectively decodes positions with low conditional entropy, thereby enabling controlled parallelism. Compared to entropy decoding, MED reduces the NFEs while maintaining model performance. Furthermore, the paper introduces reasoning-as-infilling, a framework that structures generation using a template with explicit reasoning and answer blocks. This approach enables the monitoring of answer uncertainty for early exits and facilitates the sampling of posterior reasoning traces given a known answer, which can be used for data generation and post-training.

**Strengths:**

1. The paper is well-motivated by the analysis of MDLM decoding strategies. The authors first examine both any-order and parallel decoding, revealing that left-to-right sampling remains a strong baseline and that naive parallelization substantially degrades performance. These findings provide a clear and empirical justification for developing the proposed adaptive decoding method.

2. The paper introduces the reasoning-as-infilling framework, a novel approach that structures generation by pre-filling a template with distinct reasoning and answer blocks. A key application of this framework is its ability to enable the sampling of posterior reasoning traces, which provides a more direct mechanism for data generation and model analysis compared to the conventional autoregressive models.

**Weaknesses:**

1. The paper lacks a direct empirical comparison with other acceleration techniques for MDLMs [1, 2], making it difficult to assess the competitiveness of the proposed methods. For instance, the reported 69% speed-up on GSM8K appears relatively modest.

2. Incomplete Empirical Evaluation. The paper's conclusions would be substantially strengthened by a more systematic evaluation. It is suggested to include results for MED on MATH and MBPP [3], and for the early-exit strategy on HumanEval and MBPP, to fully demonstrate the general applicability of the proposed methods.

3. The discussion of limitations could be more focused. Instead of addressing the general drawbacks of MDLMs, it would be more helpful to examine the scope of applicability and potential failure cases of the paper’s own contributions: the MED algorithm and the reasoning-as-infilling framework. This would clarify the specific strengths and limitations of the proposed approach.

4. The paper's clarity could be enhanced by rebalancing its structure. The analysis in Section 3 is extensive; condensing it would allow for a more thorough discussion of the core methods and experimental details, strengthening the paper's overall focus.

[1] Wu et al., Fast-dLLM: Training-free Acceleration of Diffusion LLM by Enabling KV Cache and Parallel Decoding.

[2] Wei et al., Accelerating Diffusion Large Language Models with SlowFast Sampling: The Three Golden Principles.

[3] Austin et al., Program Synthesis with Large Language Models.

**Questions:**

Please see Weaknesses.

---

> ### Author Response · Authors · 2025-11-24
>
> We would like to thank the reviewer for their valuable feedback.
>
> _The paper lacks a direct empirical comparison with other acceleration techniques for MDLMs [1, 2], making it difficult to assess the competitiveness of the proposed methods._
>
> Thank you for identifying the need for comparisons with other adaptive parallel decoding methods. We have run these comparison evaluations across benchmarks, and across varying sequence lengths and decoding thresholds. In several settings, at different NFE thresholds, MED outperforms both BCPD and EB, significantly reducing NFEs while preserving accuracy on benchmarks. However, there is no single best sampler. We observe that in different settings BCPD or EB-sampler are more performant, especially in very low NFE settings (but this comes with an accuracy trade-off). We have also updated our related work to include a discussion of these methods. See section A of the revised draft for the results and a detailed discussion.
>
> _Incomplete Empirical Evaluation. The paper's conclusions would be substantially strengthened by a more systematic evaluation. It is suggested to include results for MED on MATH and MBPP [3], and for the early-exit strategy on HumanEval and MBPP, to fully demonstrate the general applicability of the proposed methods._
>
> We have added MATH500 results with MED. For the early exit experiments, we focus on tasks where there are intermediate reasoning steps, such as MATH500 and GSM8k. MBPP and HumanEval are code completion tasks, generally solved without intermediate reasoning steps.
>
> _The discussion of limitations could be more focused. Instead of addressing the general drawbacks of MDLMs, it would be more helpful to examine the scope of applicability and potential failure cases of the paper’s own contributions: the MED algorithm and the reasoning-as-infilling framework. This would clarify the specific strengths and limitations of the proposed approach._
>
> We have updated the revised draft’s discussion and limitation section to include what we believe are the main limitations of our proposed methods.
>
> 1. The reasoning template requires specifying the length of the reasoning and answer blocks, however this can be mitigated by using variable-sequence length MDLM models.
> 2. The post-training methods assume that we have access to answers for questions. However, we note that this is a common assumption in finetuning papers, such as STaR and GRPO.
>
>
> _The paper's clarity could be enhanced by rebalancing its structure. The analysis in Section 3 is extensive; condensing it would allow for a more thorough discussion of the core methods and experimental details, strengthening the paper's overall focus._
>
> Thank you for the suggestion, we have condensed the analysis in section 3 in the revised draft to focus more on the proposed methods and experiments. This strengthens the paper's clarity and focus.

---

### Official Review · Reviewer_28Gt · 2025-10-31

**Soundness:** 2
**Presentation:** 2
**Contribution:** 1
**Rating:** 2
**Confidence:** 4

**Summary:**

This paper provides two techniques to make use of the parallel decoding and any-order generation attribute of MDLMs, accelerating inference: 1. predict as many tokens as possible if the entropy of their predicted distribution is low enough; 2. Set a predetermined think then answer template, the inference can exit early as soon as the entropy on the answer tokens are low enough.

**Strengths:**

1. The idea of predetermine a thinking template is reasonable. It bears similarity with <think> template in AR model, but in MDLMs we can control the thinking length.
2. The proposed method cleverly leverages the unique property of MDLMs—access to all masked token distributions—to enable adaptive parallel decoding. The theoretical grounding in KL divergence bounds adds rigor to the approach.
3. The paper provides thorough experiments across multiple benchmarks (GSM8K, Math500, HumanEval), demonstrating consistent speedups (up to 3.3×) with minimal accuracy loss. The post-hoc reasoning analysis is particularly insightful for model improvement.

**Weaknesses:**

1. The idea of fast decoding in MDLMs are already discussed in many papers. For example, Fast DLLM also decode multiple tokens according to their confidence. However, the comparisons with existing methods are ignored in this paper.
2. The template leads to human-define the length of the reasoning and answer part. I believe that for GSM8K and HumanEval, at least the answer part should be of different length. How did the authors set these hyper-parameters?
3. In table 3, it seems that we did not try accelerating a lot: the most aggressive hyper-parameter we've tried on GSM8K is still more than 0.5x steps, suggesting relatively conservative acceleration. For accelerating method, we would expect greater speedup potential.

**Questions:**

- How is Post-hoc reasoning evolved in the reasoning process actually? Does the post-hot reasoning lead to a second-time answer, and we actually evaluate with this second-time answer?

---

> ### Author Response · Authors · 2025-11-24
>
> Thank you for your valuable feedback. We have addressed your comments and included additional experiments and discussion in our revised draft, strengthening our paper.
>
> _The idea of fast decoding in MDLMs are already discussed in many papers. For example, Fast DLLM also decode multiple tokens according to their confidence. However, the comparisons with existing methods are ignored in this paper._
>
> We have now run these comparison evaluations across benchmarks, and across varying sequence lengths and decoding thresholds. In several settings, at different NFE thresholds, MED outperforms both BCPD and EB, significantly reducing NFEs while preserving accuracy on benchmarks. However, there is no single best sampler. We observe that in different settings BCPD or EB-sampler are more performant, especially in very low NFE settings (but this comes with an accuracy trade-off). We have also updated our related work to include a discussion of these methods. All methods can be combined with **reasoning-as-infilling**. See section A of the revised draft for the results and a detailed discussion.
>
>
> _The template leads to human-define the length of the reasoning and answer part. I believe that for GSM8K and HumanEval, at least the answer part should be of different length. How did the authors set these hyper-parameters?_
>
> For the early exit results in table 3, we set the length of the answer block to be fixed 10 tokens. When the answer is shorter, the model pads its answer with EOS tokens.
>
> We note that there has been concurrent work on variable length generations with MDLMs, which can obviate the need for pre-specifying the lengths of the reasoning and answer blocks. We are excited to explore this as a future direction.
>
> _In table 3, it seems that we did not try accelerating a lot: the most aggressive hyper-parameter we've tried on GSM8K is still more than 0.5x steps, suggesting relatively conservative acceleration. For accelerating method, we would expect greater speedup potential._
>
> We note that in table 3, MED and early exit achieves the following acceleration:
> 1. LLaDA GSM8k: 3.3x, 0.6% decrease in accuracy
> 2. Dream GSM8k: 2.1x, 0.8% decrease in accuracy
> 3. LLaDA MATH500: 1.9x, 1.8% decrease in accuracy
> 4. Dream MATH500: 1.8x, 0.2% decrease in accuracy
>
> These acceleration factors are achieved with a minimal loss in accuracy.
>
> We note that MED and early exits can achieve higher acceleration, however, those savings come at the expense of performance. In the revised draft, we show that MED can achieve a 5x acceleration with a 3% decrease in accuracy for MATH500 and 2% decrease in accuracy for GSM8k.
>
> We also include qualitative examples with aggressive early exits thresholds that preserve correctness.
>
> _How is Post-hoc reasoning evolved in the reasoning process actually? Does the post-hot reasoning lead to a second-time answer, and we actually evaluate with this second-time answer?_
>
> We perform post-hoc reasoning (in our case, posterior sampling given correct answers), by infilling the answer block with the ground truth answer. We then sample the reasoning block. We observe that this enables generating correct reasoning traces for many of the problems the model did not originally solve.
>
> We find that this also works for the base (non-instruction fine-tuned) model, and that fine-tuning on these posterior traces improves performance.
>
> Our posterior reasoning trace evaluations do not check for the correctness of the answer, which is provided to model. Instead, we evaluate the correctness of the reasoning steps.

---

### Official Review · Reviewer_wz9d · 2025-10-31

**Soundness:** 3
**Presentation:** 3
**Contribution:** 2
**Rating:** 4
**Confidence:** 4

**Summary:**

This paper reexamines MDLMs and focuses on two widely claimed advantages: multi-token decoding and any-order decoding. The authors propose Multi-token Entropy Decoding (MED), an adaptive sampler that achieves 2–3× faster inference without accuracy loss, and a Reasoning-as-Infilling framework that enables structured reasoning, early exits, and posterior sampling of reasoning traces. Extensive experiments validate the effectiveness of the proposed methods.

**Strengths:**

1. Multi-token Entropy Decoding (MED) is theoretically grounded and empirically validated. Its entropy-based criterion provides a principled way to control decoding errors, achieving 2–3× faster inference without accuracy loss.


2. The Reasoning-as-Infilling framework reveals a novel and previously unexplored property of MDLMs: generating correct reasoning traces for 43% of problems originally solved incorrectly.

**Weaknesses:**

The paper lacks sufficient discussion and comparison with closely related works. Specifically, [1] introduces a confidence-threshold-based sampling approach, and [2] proposes an entropy-bounded unmasking procedure. Both methods are conceptually similar to the proposed Multi-token Entropy Decoding (MED) and also provide theoretical analyses. Unfortunately, the authors do not discuss these works in depth or include empirical comparisons with them.

[1] Wu et al. Fast-dLLM: Training-free Acceleration of Diffusion LLM by Enabling KV Cache and Parallel Decoding. arXiv 2025.05.

[2] Ben-Hamu et al. Accelerated Sampling from Masked Diffusion Models via Entropy Bounded Unmasking. arXiv 2025.05.

**Questions:**

1. Could the proposed Early Exit mechanism terminate decoding prematurely—i.e., when the final answer has been generated but the reasoning chain is still incomplete?

2. The Reasoning-as-Infilling framework shows that MDLMs can generate correct reasoning traces for problems originally solved incorrectly. What insights does this provide for improving MDLM training or architecture design in the future?

---

> ### Author Response · Authors · 2025-11-24
>
> We would like to thank you for your thoughtful feedback. We have included additional experiments and comparisons to address the concerns that your raised. These new results significantly strengthen the quality of our work.
>
> _Could the proposed Early Exit mechanism terminate decoding prematurely—i.e., when the final answer has been generated but the reasoning chain is still incomplete?_
>
> Yes, this is one of the findings of the paper. In several cases, the model was certain about the answer blocks before filling the complete reasoning block. In table 4, we note that early exits skip 24% of the reasoning tokens and directly generate the answer with a 0.8% loss in accuracy.
>
> In section F of the revised draft, we provide sample outputs with early exits at varying thresholds.
>
> We compare:
> 1. Sampling from the model with no reasoning template and no early exit.
> 2. Reasoning template with no early exit.
> 3. Reasoning template with early exits with varying thresholds.
> 4. Reasoning template with the model producing the answer directly with no reasoning tokens decoded.
>
> We observe that in both (1) and (2), the model decodes all 256 tokens to produce the correct answer. When the model is forced to answer without any reasoning, it produces an incorrect answer. However, with early exits we show that even <10 reasoning tokens are sufficient for the model to converge on the correct answer.
>
> > **LLaDA (No reasoning template, No Early Exits) [Correct]:**
> >
> > ```text
> > To determine how much Janet makes every day at the farmers' market, we need to follow these steps:
> > 1. Calculate the total number of eggs laid by Janet's ducks per day.
> > 2. Subtract the number of eggs Janet eats and bakes for her friends from the total number of eggs laid.
> > 3. Determine the number of eggs she sells at the farmers' market.
> > 4. Calculate the total revenue from selling the eggs.
> > Step 1: Janet's ducks lay 16 eggs per day.
> > Step 2: Janet eats 3 eggs for breakfast every morning and bakes 4 eggs for her friends every day. Therefore, she eats and bakes a total of \(3 + 4 = 7\) eggs per day.
> > Step 3: The number of eggs she sells at the farmers' market is \(16 - 7 = 9\) eggs per day.
> > Step 4: Janet sells each egg for $2. Therefore, the total revenue from selling 9 eggs per day is \(9 \times 2 = 18\) dollars.
> > So, the amount Janet makes every day at the farmers' market is \(\boxed{18}\).
> > ```
>
>
> > **LLaDA output (Reasoning template, Early Exit γ = 0.5) [Correct]:**
> > ```
> > To determine how much Janet makes every day at the farmers’ market, we need to follow these steps:
> > Step 1: Calculate the total number of eggs laid by Janet’s ducks
> > per day.
> > <|mdm_mask|>…<|mdm_mask|> per<|mdm_mask|>…<|mdm_mask|>
> > The answer is \boxed{18}.
> > ```
>
>
> > **LLaDA output (Reasoning template, Forced Answer) [Incorrect]:**
> >```
> >|mdm_mask|>...<|mdm_mask|>. The answer is \boxed{14}.
> > ```
>
>
> _The Reasoning-as-Infilling framework shows that MDLMs can generate correct reasoning traces for problems originally solved incorrectly. What insights does this provide for improving MDLM training or architecture design in the future?_
>
> Thank you for this question. We have added new results showing that fine-tuning on the posterior generated reasoning traces improves model performance. We use the GSM8k training data that consists of (question, reasoning, answer) triplets. We then generate posterior reasoning traces given the question and the answer.
>
> We find that low rank fine-tuning on the non-instruct finetuned model generated reasoning traces provided a significant improvement (51.2% -> 66.1%) over the base model performance. As a benchmark, we also show that fine-tuning on the posterior generated reasoning traces provides a similar improvement (51.2% -> 64.6%)  as fine-tuning on the high-quality human annotated GSM8k reasoning traces.
>
> This experiment demonstrates that reasoning-as-infilling with MDLMs provides a source of high-quality data generation for post-training.
>
> We also show in Figure 5 that MDLMs can score their own posterior reason chains, and that lower quality post-hoc reasoning could be filtered.
>
>
> _The paper lacks sufficient discussion and comparison with closely related works. Specifically, [1] introduces a confidence-threshold-based sampling approach, and [2] proposes an entropy-bounded unmasking procedure._
>
> Thank you for your question. We have run these comparison evaluations across benchmarks, and across varying sequence lengths and decoding thresholds. In several settings, at different NFE thresholds, MED outperforms both BCPD and EB, significantly reducing NFEs while preserving accuracy on benchmarks. However, there is no single best sampler. We observe that in different settings BCPD or EB-sampler are more performant, especially in very low NFE settings (but this comes with an accuracy trade-off). We have also updated our related work to include a discussion of these methods. See section A of the revised draft for the results and a discussion.

---

### Official Review · Reviewer_Buvt · 2025-10-31

**Soundness:** 2
**Presentation:** 2
**Contribution:** 2
**Rating:** 4
**Confidence:** 4

**Summary:**

This paper investigates various inference aspects and their benefits for masked diffusion models. The authors first examine two commonly discussed advantages: any-order sampling and multi-token prediction. They find that any-order sampling offers limited gains, while multi-token prediction degrades performance. To address this, the authors propose a Multi-token Entropy Decoding (MED) strategy, which dynamically determines the number of tokens to decode at each inference step, resulting in an inference-time speedup. They further demonstrate additional efficiency improvements through early exiting based on the model’s uncertainty in its answers.

**Strengths:**

The proposed method MED with early exist seems to provide good inference efficiency gains on coding and reasoning tasks.

**Weaknesses:**

- Several claims in the paper are not sufficiently supported by experimental evidence and appear somewhat overstated.
     - For instance, to support the claim that “MDLM reasoning posterior yields high-quality traces,” the authors only provide correctness scores judged by Qwen and GPT. However, correctness alone does not offer a complete view of trace quality. It would also be more informative to include correctness scores for a baseline model, such as LLaMA 3.1 (or another model with comparable capabilities), to better contextualize the benefits of MDLM’s reasoning quality relative to other autoregressive models.
     - The claim that left-to-right sampling performs on par with any-order sampling also seems somewhat overstated and comes with a few caveats. Except for the Dream 7B model on the HumanEval task, any-order sampling with certain block sizes consistently outperforms left-to-right sampling. It is also worth noting that the Dream 7B model was adapted from an autoregressive model, which naturally introduces a left-to-right inductive bias. Additionally, tasks like GSM8K are known to rely on predominantly left-to-right reasoning patterns (see [1]). The true advantage of any-order sampling is more likely to appear in tasks requiring non-linear reasoning (e.g., Sudoku, though admittedly more synthetic in nature).

- It would also be more informative for readers if all comparisons were presented in terms of Number of Function Evaluations (NFEs) rather than the number of parallel tokens decoded (as in Tables 2 and 3). The number of parallel tokens directly reduces NFEs by a proportional factor, which results in a major change, whereas reducing the number of inference steps in diffusion impacts NFEs without such drastic changes. For instance, it is unclear how the entropy decoding method operates with 96 or 112 NFEs in Table 3.

- Finally, the paper should include a more comprehensive comparison with the EB sampler and generate_until sampler from Ben-Hamu et al. I would also like to note that the Ben-Hamu et al. paper appeared on arXiv at the end of May, and according to ICLR’s submission guidelines, papers released after July 1 are considered concurrent work.

[1] Premise Order Matters in Reasoning with Large Language Models. Chen et al. 2024

**Questions:**

- It would be interesting to understand the performance left-to-right decoding and any order decoding with different block sizes work on R-GSM8k proposed in [1]
- Why is HumanEval on Llada with left-to-right accuracy 15.24 in Table 1 and 11.0% in Table 7? Typo?

---

> ### Author Response · Authors · 2025-11-24
>
> We would like to thank you for your valuable comments and time. We have addressed each of the points that you made and added clarifications below. Incorporating your feedback has significantly strengthened our work.
>
> _To support the claim that “MDLM reasoning posterior yields high-quality traces,” the authors only provide correctness scores judged by Qwen and GPT. However, correctness alone does not offer a complete view of trace quality. It would also be more informative to include correctness scores for a baseline model, such as LLaMA 3.1 to better contextualize the benefits of MDLM’s reasoning quality relative to other autoregressive models._
>
> Thank you for this question. We have added this comparison to the revised draft.
>
> We analyze the questions solved incorrectly by Llama-3-8B. For these questions, we then prompt Llama with the question and the answer as a hint (i.e. STaR). In the table below, we observe that the **posterior reasoning traces with LLaDA score higher than Llama generated STaR traces.**
>
> | **Model**                      | **Qwen-PRM** | **GPT-4o** |
> |--------------------------------|--------------|------------|
> | Llama Instruct *STaR* (L = 256)         | 0.39         | 0.33       |
> | Llama Instruct *STaR* (L = 512)         | 0.39         | 0.40       |
> |--------------------------------|--------------|------------|
> | LLaDA Base (L = 128)           | 0.34         | 0.41       |
> | **LLaDA Instruct (L = 128)**   | **0.46**     | **0.52**   |
>
>
>
>
> We find that reasoning-as-infilling enables sampling high-quality reasoning chains from **even the non-instruct tuned model.**
>
> Going further, we **fine-tuned the LLaDA non-instruction fine-tuned model on its posterior reasoning traces** for the GSM8k dataset. We observe that fine-tuning on the posterior generated reasoning traces (with LoRA, and a limited compute budget) improves test performance. As a benchmark, we fine-tune the base model on the GSM8k human-annotated reasoning traces which provide a similar (but smaller) gain.
>
> This experiment provides evidence that reasoning-as-infilling provides a source of high-quality data for post-training.
>
>
>
> | **Model**                                    | **Post-training Data**                                              | **GSM8K Test Acc.** |
> |----------------------------------------------|----------------------------------------------------------------------|----------------------|
> | LLaDA 8B-Base (No reasoning template)        | –                                                                    | **13.9%**           |
> | LLaDA 8B-Base (With reasoning template)      | –                                                                    | **51.2%**           |
> | **Finetuned\*** LLaDA 8B-Base                 | GSM8K *(c, r_gold, a_gold)* *(n = 7473)*                             | **64.6% (+13.4%)**  |
> | **Finetuned\*** LLaDA 8B-Base                 | GSM8K Posterior *(c, r_posterior, a_gold)* *(n = 7473)*             | **66.1% (+14.9%)**  |
> | LLaDA-8B Instruct                             | Misc. Instruction Data *(n = 4.5 million)*                          | **75.96%**          |
>
> _The claim that left-to-right sampling performs on par with any-order sampling also seems somewhat overstated and comes with a few caveats._
>
>
>
>
> Thank you for this question. We do not claim that left-to-right decoding is superior to any-order decoding for arbitrary problems.  In table 1 of the previous draft, we observe that on puzzle solving tasks, such as sudoku, there are benefits to any-order sampling. However, for several problems requiring reasoning such as math and coding, left-to-right decoding outperforms purely any-order decoding.
>
> Furthermore, in appendix A, we observe that the most performant block sampling approaches behave highly auto-regressively. For example, we observe that even for block 32 methods, ~50% of the tokens are generated autoregressively, and the average token is ~3 positions from the left-most position.
>
> We do not claim that the ability to do any-order decoding is not useful, however, we highlight that there are several other benefits of MDLMs besides any-order decoding. In fact, we use the ability of MDLMs to in-fill and any-order decode to generate posterior reasoning traces given answers.

---

> > ### Author Response · Authors · 2025-11-24
> >
> > _It would be interesting to understand the performance left-to-right decoding and any order decoding with different block sizes work on R-GSM8k proposed in [1]_
> >
> > We cannot locate any database, codebase or pseudo-code related to the R-GSM dataset or creating the dataset. However, this is an exciting direction for future work!
> >
> >
> > _It would also be more informative for readers if all comparisons were presented in terms of Number of Function Evaluations (NFEs) rather than the number of parallel tokens decoded (as in Tables 2 and 3)._
> >
> > Table 3 (in the original draft) has a column labeled NFEs which refers to the number of function evaluations. For MED, we set the sequence length as the ceiling on the number of steps. However, each step in MED can decode multiple tokens, therefore, MED typically requires fewer steps than specified.
> >
> >
> > Table 2 studies the impact of reducing the number of inference steps by decoding k tokens in parallel. In table 2, we generate a sequence of length 128 and decode e k=1, 2, 4 tokens in each step, this results in 128, 64, 32 NFEs respectively.
> >
> >
> >
> > _The paper should include a more comprehensive comparison with the EB sampler and generate_until sampler from Ben-Hamu et al._
> >
> > We have run these comparison evaluations across benchmarks, and across varying sequence lengths and decoding thresholds. In several settings, at different NFE thresholds, MED outperforms both BCPD and EB, significantly reducing NFEs while preserving accuracy on benchmarks. However, there is no single best sampler. We observe that in different settings BCPD or EB-sampler are more performant, especially in very low NFE settings (but this comes with an accuracy trade-off). We have also updated our related work to include a discussion of these methods. See section A of the revised draft for the results and a detailed discussion.
> >
> >
> > _Why is HumanEval on Llada with left-to-right accuracy 15.24 in Table 1 and 11.0% in Table 7? Typo?_
> >
> > Thank you for catching this, we have fixed the typo. Note that these results are for zero-shot performance.

---

### Author Response · Authors · 2025-11-24
**General Reply**

We would like to thank our reviewers for their thoughtful and constructive feedback. Below, we include new results and comparisons to address their concerns. At the reviewers' request, we have included additional reasoning-as-infilling experiments and comparisons with multi-decoding methods. These findings strengthen and clarify the contributions of our work.


The most significant aspect of our work is the **reasoning-as-infilling** framework, and resulting benefits.  Discussions around MDLMs focus on any-order and parallel token decoding. However, we demonstrate that the infilling capabilities of MDLMs unlock new prompting approaches. We focus on reasoning-as-infilling, where we prefill a *reasoning template* that diffentiates between reasoning and answer positions. This enables:
1. *Early exits,* where if the model is certain about the answer, we then skip the remaining reasoning steps. This leads to a 3.3× speed when combined with multi-token entropy decoding (MED).
2. *Post-hoc reasoning*, where given question-answer pairs, we generate reasoning traces conditioned on the answer. On the GSM8k dataset, we find that supervised fine-tuning on these reasoning traces can improve the model more than supervised fine-tuning on the human-annotated GSM8k reasoning traces.

3. *Scoring reasoning traces*, where given an answer, we can score the reasoning process for correctness at intermediate steps using the distributions of the answer block, without an external verifier or roll-outs. These scores correlate with whether the reasoning steps lead to a correct answer.
Below, we include additional results and clarifications that address each point raised by our reviewers. We have also **uploaded a revised draft** that incorporates their feedback and the new results.


### 1. Early Exits

MDLMs enable computing an upper-bound on the entropy of the answer block at intermediate sampling steps. During reasoning, when the entropy of the answer block falls below a specified threshold, we can exit the reasoning block and generate an answer. Early exits can accelerate inference while preserving accuracy (**Updated examples in section F of the revised draft and below**)

Reviewers asked about the qualitative behavior of early exits. We observe that for several problems, early exits can terminate with the correct answer after decoding only a few reasoning tokens. We include examples in the appendix.


> **LLaDA output (Reasoning template, Early Exit γ = 0.5) [Correct]:**
> ```
> To determine how much Janet makes every day at the farmers’ market, we need to follow these steps:
> Step 1: Calculate the total number of eggs laid by Janet’s ducks
> per day.
> <|mdm_mask|>…<|mdm_mask|> per<|mdm_mask|>…<|mdm_mask|>
> The answer is \boxed{18}.
> ```


> **LLaDA output (Reasoning template, Forced Answer) [Incorrect]:**
>```
>|mdm_mask|>...<|mdm_mask|>. The answer is \boxed{14}.
> ```

### 2. Post-hoc reasoning elicits correct reasoning chains from MDLMs, and these data are useful for post-training  (*Updated Results*)
By pre-filling the answer block with the correct answer, MDLMs offer a method for sampling reasoning traces from the posterior distribution of the reasoning trace conditioned on the question and the answer. We evaluate the correctness of the reasoning steps in the posterior reasoning chains generated for problems that the model failed to solve using the Qwen process reward model and GPT-4o.


- **Update #1:**  In the revised draft, we show that we can even generate post-hoc reasoning data using non-instruction fine-tuned models. Here, we observe that prefilling a reasoning template and answer with the non-instruction finetuned LLaDA model results in correct reasoning traces for 36% of questions that were initially answered incorrectly. This demonstrates that reasoning-as-infilling can elicit high-quality reasoning chains, even for the non-instruction fine-tuned model.

    | **Model**        | **Qwen-PRM** | **GPT-4o** |
    |------------------|--------------|------------|
    | LLaDA Base       | _0.31_         | _0.36_       |
    | LLaDA Instruct   | 0.38         | 0.43       |


- **Update #2:**  As requested by Reviewer Buvt, we also examine the problems LLama fails to solve. Here we find that LLaDA posterior reasoning **outperforms prompting Llama models with answer hints (i.e. STaR).**

    | **Model**                      | **Qwen-PRM** | **GPT-4o** |
    |--------------------------------|--------------|------------|
    | Llama3-8B Instruct *STaR* (Seq. Length = 256)         | 0.39         | 0.33       |
    | Llama3-8B Instruct *STaR* (Seq. Length = 512)         | 0.39         | 0.40       ||
    | LLaDA Base (Seq. Length = 128)           | 0.34         | 0.41       |
    | **LLaDA Instruct (Seq. Length = 128)**   | **0.46**     | **0.52**   |

---

> ### Author Response · Authors · 2025-11-24
>
> - **Update #3:** In the revised draft, we also show that fine-tuning LLaDA on this posterior data improves performance, see updated results below and tables 3,4,5 of the revised draft. We found that parameter-efficient fine-tuning on posterior traces, even with a limited compute budget, can improve base model performance on GSM8k just as much as supervised fine-tuning on the human-annotated GSM8k gold reasoning traces.
>
>     | **Model**                                    | **Post-training Data**                                              | **GSM8K Test Acc.** |
>     |----------------------------------------------|----------------------------------------------------------------------|----------------------|
>     | LLaDA 8B-Base (No reasoning template)        | –                                                                    | **13.9%**           |
>     | LLaDA 8B-Base (With reasoning template)      | –                                                                    | **51.2%**           |
>     | **Finetuned\*** LLaDA 8B-Base                 | GSM8K *(q, r_gold, a_gold)* *(n = 7473)*                             | **64.6% (+13.4%)**  |
>     | **Finetuned\*** LLaDA 8B-Base                 | GSM8K Posterior *(q, r_posterior, a_gold)* *(n = 7473)*             | **66.1% (+14.9%)**  |
>     | LLaDA-8B Instruct                             | Misc. Instruction Data *(n = 4.5 million)*                          | **75.96%**          |
>
> ### 3. Reasoning-as-Infilling enables scoring reasoning chains at intermediate steps
>
> Given access to an answer, the marginal distributions produced by the MDLMs enable scoring reasoning at intermediate steps during post-training. This could enable steering the model toward correct reasoning chains during post-training. We observe that the probabilities of answer tokens in a gold completion is more predictive of final correctness than an external PRM model, see figure 3.
>
> **Update:** Notably, we also find that measuring answer block log probabilities even enables filtering lower-quality reasoning chains generated via posterior sampling. We can unmask a posterior reasoning chain without the prefilled answer and the MDLM answer probabilities are predictive of GPT4-o reasoning chain correctness scores (AUC = 0.74, see figure 6).
>
> ## Multi-token Entropy Decoding Baseline Comparison (*Updated Results*)
>
> In addition to the reasoning-as-infilling method, we also propose Multi-Token Entropy Decoding (MED), an adaptive sampler that decodes tokens in parallel based on entropy thresholds. Reviewers asked us to compare to two parallel decoding algorithms: Entropy-Bound Sampler (EB) and Blockwise confidence-aware parallel decoding (BCPD).  We deeply appreciate the encouragement to include these comparisons, which strengthen the quality of our paper.
>
> We have run these comparison evaluations across benchmarks, and across varying sequence lengths and decoding thresholds. In several settings, at different NFE thresholds, MED outperforms both BCPD and EB, significantly reducing NFEs while preserving accuracy on benchmarks. However, there is no single best sampler. We observe that in different settings BCPD or EB-sampler are more performant, especially in very low NFE settings (but this comes with an accuracy trade-off). We have also updated our related work to include a discussion of these methods.
>
> Our full set of results indicate that MED is one of several possible parallel samplers, but a useful addition to the dLLM sampling toolkit. Notably MED has several advantages over BCPD, which requires greedy decoding. In contrast, MED supports any probabilistic sampling algorithm, and can be used for multi-particle sampling methods such as best-of-N, sequential Monte-Carlo, and post-training methods, such as GRPO, that require multiple samples for each query.
>
> Notably, all parallel samplers could be combined with the proposed **reasoning-as-infilling** approach.

---

### Author Response · Authors · 2025-12-04
**Note to Area Chair (1/2)**

1. Overview
2. Positive Reviewer Comments
3. Shared Reviewer Concern and *How We Addressed This*
4. Specific Reviewer Concerns and *How We Addressed Them*

## 1) Overview


**What are the benefits of MDLMs?** In our work, we first observe that, despite their any-order decoding capacity, Masked Diffusion Language Models (MDLMs) behave highly autoregressively on reasoning and coding tasks, sampling a large portion of their tokens left-to-right. Additionally, we show that fixed parallel sampling, even decoding two tokens at a time, severely hurts performance.

As a result, the focus of this paper is on re-examining the inference-time and post-training benefits of MDLMs. We propose:
- *Reasoning/prompting-as-infilling.* Here we make use of the in-filling capability of MDLMs. Rather than prompting MDLMs with just an instruction prefix, we prefill a reasoning template that distinguishes between **reasoning** and **answer** blocks, and then infill the template with the MDLM. This enables:

    - **Early exits based on answer block uncertainty**. We find that diffusion models often know the answer early, *before* finishing their chain-of-thought.
    - **Post-hoc reasoning traces**. Given question-answer pairs, we can prefill the answer block and then sample reasoning traces conditioned on the answer. We incude an extensive evaluation of these post-hoc reasoning traces, showing that these traces produce mathematically correct reasoning steps for questions that the model initially fails to solve. We also demonstrate that finetuning on these post-hoc reasoning traces improves model performance.
    - **Eliciting correct reasoning chains from a base (non-instruction-tuned) MDLM.** Templates enable controlling the format of model outputs and budgeting compute, even from the base model.
    - **MDLMs provide internal process rewards for post-training**: Given question-answer pairs, we show that MDLMs can score the correctness of the reasoning process using the distributions of the answer block. We also compare these internal PRM scores with a trained Qwen-PRM, 7.5B model. Our experiments demonstrate the internal PRM score provided by the MDLM model has higher correlation with correctness than the 7.5B model.

- *Multi-token Entropy Decoding (MED)*: To further accelerate inference while preserving accuracy, we also propose an adaptive parallel decoding approach, **MED**, which is competitive with recent parallel decoding methods, and even yields higher accuracies at several compute settings.


## 2) Positive Reviewer Feedback

Reviewer wz9d: "The Reasoning-as-Infilling framework reveals a novel and previously unexplored property of MDLMs"

Reviewer 28Gt: "The proposed method cleverly leverages the unique property of MDLMs"

Reviewer wz9d: "Reasoning-as-Infilling framework that enables structured reasoning, early exits, and posterior sampling of reasoning traces. Extensive experiments validate the effectiveness of the proposed methods"


Reviewer s1QC: "This approach enables the monitoring of answer uncertainty for early exits and facilitates the sampling of posterior reasoning traces given a known answer, which can be used for data generation and post-training."

Reviewer 28Gt: "The paper provides thorough experiments across multiple benchmarks (GSM8K, Math500, HumanEval), demonstrating consistent speedups (up to 3.3×) with minimal accuracy loss."


## 3) Shared Reviewer Concern


All reviewers focused on the _same key concern_: In one part of our paper, we propose an adaptive parallel token decoding method based on model entropies, MED. All reviewers stated that the weakness of our paper was a missing comparison between our proposed method and two recent parallel decoding methods. **During the rebuttal process, we addressed these concerns by running these baselines and including comparisons in our paper** (See figure 4 of the revised draft).

In several settings, at different NFE thresholds, MED outperforms both alternatives, significantly reducing NFEs while preserving accuracy on benchmarks. However, there is no single best sampler. We observe that in different settings BCPD or EB-sampler are more performant, especially in very low NFE settings (but this comes with an accuracy trade-off). We have also updated our related work to include a discussion of these methods.

Addressing this constructive feedback has significantly strengthened our paper.

---

> ### Author Response · Authors · 2025-12-04
> **Note to Area Chair (2/2)**
>
> ## 4) Specific Reviewer Concerns
>
> ### Reviewer Buvt
>
> 1. **Concern:** Comparisons with recent parallel decoding approaches.
>  **Response:** Addressed. See the Shared Reviewer Concern section.
>
> 2. **Concern**: It would also be more informative to include correctness scores for a baseline model, such as LLaMA 3.1.
> **Response:**  Included **comparisons of MDLM posterior sampling to Llama STaR**, and find that MDLM posterior traces yield higher scores. We also further elaborated on our reasoning chain evaluations, and introduced new fine-tuning experiments that utilize post-hoc data. These fine-tuning experiments demonstrate that the posterior traces sampled from MDLMs can provide useful training data.
>
> 3. **Concern:** The claim that left-to-right sampling performs on par with any-order sampling also seems somewhat overstated and comes with a few caveats.
> **Response:** Clarified that while there is an accuracy improvement from block any-order sampling for several tasks, *purely any-order decoding performs significantly worse than left-to-right decoding*. Furthermore, we study the resulting behavior of top performing block sampling approaches for math and code, and find that these methods sample ~50% of their tokens left-to-right, and on average ~3 positions from the left.
>
>
> ### Reviewer wz9d
>
>  1. **Concern:** Comparisons with recent parallel decoding approaches.
>  **Response:** Addressed. See the Shared Reviewer Concern section.
>
> 2. **Question**: What insights does post-hoc reasoning provide for improving MDLM training?
> **Response:** Added results for *post-hoc fine-tuning experiments*, which show that fine-tuning on the posterior generated traces improves model performance.
>
> ### Reviewer 28Gt
>  1. **Concern:** Comparisons with recent parallel decoding approaches.
>  **Response:** Addressed. See the Shared Reviewer Concern section.
>
> 2. **Concern**: The reasoning template requires specifying the length of the reasoning and answer blocks. How are hyperparameters selected?
> **Response**: We have provided clarifications about these hyper-parameter choices in the *reasoning-as-infilling* and *post-hoc reasoning* approaches. Notably, longer answer blocks can be padded with EOS tokens. We also note how variable length models obviate can the need to choose specific lengths.
>
> 3. **Question** How are the post-hoc reasoning chains evaluated? **Response**: Added clarifications regarding the post-hoc reasoning chain evaluations (including fine-tuning results, see [link](https://openreview.net/forum?id=2mkYB5KUwa&noteId=6IoPsUc0Pb)).
>
>
> ### Reviewer s1QC
>
>  1. **Concern:** Comparisons with recent parallel decoding approaches.
>  **Response:** Addressed. See the Shared Reviewer Concern section.
>
> 2. **Concern:** The paper's clarity could be enhanced by rebalancing its structure and expanding the limitations section.
> **Response**: We have adjusted the limitations section to focus on reasoning-as-infilling and shortened the analysis of the benefits of the any-order and fixed parallel token decoding in section

---

### Meta-Review · Area_Chair_TCK7 · 2026-01-03

**Summary:**

This paper studies masked diffusion language models (MDLMs) for reasoning and coding. It questions whether commonly claimed decoding advantages, such as any-order or fixed multi-token decoding, actually help on reasoning tasks, and proposes two ideas: framing reasoning as an infilling problem with early exit, and a multi-token entropy-based decoding strategy to adapt the degree of parallelism. Reviewers found the topic timely and the direction interesting, but there are concerns about limited novelty relative to closely related recent methods, uneven and selective empirical evaluation, and claims that are stronger than what the evidence consistently supports across tasks.

**Reviewer Concerns:**

### Concerns addressed by the rebuttal ###

- The rebuttal adds comparisons to several recent parallel or entropy-based decoding methods, which addresses a major gap noted by multiple reviewers.

- The authors provide additional baselines for evaluating post-hoc reasoning traces and clarify parts of the experimental setup, including template design and decoding details.

- The framing around any-order decoding is softened and clarified, acknowledging cases where left-to-right or block-structured decoding performs better.

### Concerns that remain outstanding ###

- Despite added comparisons, the core contribution still feels incremental relative to concurrent work, and the paper does not clearly establish when or why the proposed approach is preferable beyond specific settings.

- The empirical evaluation remains selective. While some new results are added, reviewers expected a more systematic study across reasoning and coding tasks, especially to support claims about early exit and efficiency.

- The quality and usefulness of generated reasoning traces are still only partially characterized, relying mostly on correctness-based proxies rather than deeper analysis of faithfulness or failure modes.

- Some aspects of the method assume access to correct answers during post-processing, which limits applicability and weakens the generality of the claims.

**Reviewer Scores:**

- Reviewer Buvt: 4 -> 4. While the rebuttal addresses missing baselines and clarifies several points, the reviewer’s concerns about overstated claims and overall contribution are unlikely to be fully resolved.

- Reviewer wz9d: 4 -> 6. The added comparisons and clarifications directly address this reviewer’s main concerns, likely moving the assessment to borderline positive.

- Reviewer 28Gt: 2 -> 4. Clarifications and additional experiments may alleviate some issues, but concerns about novelty and evaluation breadth likely keep the score below the acceptance threshold.

- Reviewer s1QC: 2 -> 4. Improvements to comparisons and structure help, but the lack of a comprehensive evaluation across tasks remains a significant concern.

---

### Decision · Program_Chairs · 2026-01-26

Reject